

# Population genetics of the freshwater fish *Prochilodus magdalenae* (Characiformes: Prochilodontidae), using species-specific microsatellite loci

Ricardo M. Landínez-García[1], Juan Carlos Narváez[2] and
Edna J. Márquez[1]

[1] Facultad de Ciencias, Escuela de Biociencias, Laboratorio de Biología Molecular y Celular, Universidad Nacional de Colombia—Sede Medellín, Medellín, Colombia
[2] Grupo de Biodiversidad y Ecología Aplicada, Laboratorio de Genética Molecular, Universidad del Magdalena, Santa Marta, Magdalena, Colombia

Corresponding author
Edna J. Márquez,
ejmarque@unal.edu.co

## ABSTRACT

*Prochilodus magdalenae* is a freshwater fish endemic to the Colombian Magdalena-Cauca and Caribbean hydrographic basins. The genetic structure patterns of populations of different members of *Prochilodus* and the historic restocking of its depleted natural populations suggest that *P. magdalenae* exhibits genetic stocks that coexist and co-migrate throughout the rivers Magdalena, Cauca, Cesar, Sinú and Atrato. To test this hypothesis and explore the levels of genetic diversity and population demography of 725 samples of *P. magdalenae* from the studied rivers, we developed a set of 11 species-specific microsatellite loci using next-generation sequencing, bioinformatics, and experimental tests of the levels of diversity of the microsatellite loci. The results evidenced that *P. magdalenae* exhibits high genetic diversity, significant inbreeding coefficient ranging from 0.162 to 0.202, and signs of erosion of the genetic pool. Additionally, the population genetic structure constitutes a mixture of genetic stocks heterogeneously distributed along the studied rivers, and moreover, a highly divergent genetic stock was detected in Chucurí, Puerto Berrío and Palagua that may result from restocking practices. This study provides molecular tools and a wide framework regarding the genetic diversity and structure of *P. magdalenae*, which is crucial to complement its baseline information, diagnosis and monitoring of populations, and to support the implementation of adequate regulation, management, and conservation policies.

## INTRODUCTION

The family Prochilodontidae (Teleostei: Characiformes) comprises the genera *Prochilodus*, *Semaprochilodus*, and *Ichthyoelephas*, and encompasses 21 Neotropical freshwater fish species in the main river basins of South America (*Castro & Vari, 2004*). Most of the prochilodontids exhibit large body sizes, high fecundities, and abundances, representing around 50–80% of the biomass caught by the subsistence and commercial fisheries in some

regions of their distribution area (*Barroca et al., 2012b*; *Melo et al., 2016a*). Furthermore, some members of Prochilodontidae constitute a potential resource for fish farming due to certain characteristics such as their fast growth and weight increase, rustic management, and high economic value (*Flores-Nava & Brown, 2010*; *DellaRosa et al., 2014*; *Roux et al., 2015*).

In addition to the economic importance, Prochilodontidae plays an important trophic role in aquatic ecosystems. These detritivorous and migratory fishes contribute to the nutrient cycling, distribution, equilibrium, and maintenance of energetic flows and support a wide trophic network for a great number of predators (*Flecker, 1996*). Hence, the adequate management of fisheries is crucial for the maintenance of high productivity and permanent resource availability, as well as to guarantee the stability and continuity of the aquatic ecosystems (*Taylor, Flecker & Hall, 2006*; *Batista & Lima, 2010*).

The bocachico *Prochilodus magdalenae* Steindachner 1878 is the most representative endemic species of the Colombian ichthyofauna, considered the emblematic fishery resource of the Magdalena-Cauca Basin, with an estimated unload for the Magdalena Basin of 2,182.67 metric tons in 2013 (Colombian fishing statistical service: SEPEC). However, between 1978 and 2012, this species experienced drastic decreases in its population densities, catches (approx. 85%), and mean catch sizes. These effects resulted from overfishing during migratory periods, violations of legislation related to mean catch sizes, and habitat disturbances including deforesting, floodplain lake desiccations, agrochemical or chemical contamination derived from farming and mining activities, sedimentation, and dam/hydropower construction, among others (*Cortés Millán, 2003*; *Lasso et al., 2010*; *Mojica et al., 2012*).

To counteract this detrimental situation, several state regulations were implemented for the management and conservation of *P. magdalenae* (*Usma et al., 2009*; *Lasso et al., 2010*; *Mojica et al., 2012*). Specifically, this fish resource was cataloged as critically endangered in 2002 and as vulnerable since 2012 in the Colombian Red List of freshwater fishes (*Mojica et al., 2012*). Additionally, national regulations of territorial entities and autonomous corporations focused their efforts on the restocking of natural stocks threatened in the last 20 years (INPA regulation 531-1995; ANLA, INCODER, AUNAP regulation 2838-2017). However, these last-mentioned activities are not based on knowledge of the population genetics of *P. magdalenae* and their ecological, genetic, and sanitary impacts are unknown due to the lack of programmatic monitoring and regulation of fish farming (*Povh et al., 2008*; *FAO, 2011*).

Moreover, population genetic studies of *P. magdalenae* are scarce, and fragmented (*López-Macías et al., 2009*; *Aguirre-Pabón, Narváez-Barandica & Castro-García, 2013*; see *Mancera-Rodríguez, Márquez & Hurtado-Alarcón, 2013*; *Orozco-Berdugo & Narváez-Barandica, 2014*; *Hernández, Navarro & Muñoz, 2017*), and most of the required information regarding the origin, genetic diversity and structure of juveniles used for restocking of natural stocks remains unavailable. Hence, natural stocks of *P. magdalenae* are highly susceptible to experiencing disturbances of their genetic background resulting

from artificial mixtures of genetic stocks with different evolutionary histories or, alternatively, from the high competition for resources among different stocks.

Since *P. magdalenae* performs long longitudinal migrations (ca. 1,224 km; velocity: 55.6 km/day) (*López-Casas et al., 2016*), it is reasonable to think that its natural stocks experience extensive gene flow. However, the observation that *Prochilodus lineatus* (*Godoy, 1959*) and *Prochilodus argenteus* (*Godinho & Kynard, 2006*) show fidelity to spawning sites ("homing") suggests that *P. magdalenae* may exhibit a population genetic structure even in the absence of physical barriers.

Furthermore, previous genetic studies have found the population structure and/or coexistence of multiples stocks along the Magdalena River and several of its tributaries (*López-Macías et al., 2009*; see *Mancera-Rodríguez, Márquez & Hurtado-Alarcón, 2013*; *Orozco-Berdugo & Narváez-Barandica, 2014*). Although this structure may result from the unregulated restocking of the natural stocks, it could also reflect a natural behavior of *P. magdalenae* since similar patterns of genetic population structure have been found in other congeners such as *P. reticulatus* (*López-Macías et al., 2009*), *P. argenteus* (*Hatanaka & Galetti, 2003*; *Hatanaka, Henrique-Silva & Galetti, 2006*; *Barroca et al., 2012a*), *P. lineatus* (*Ramella et al., 2006*; *Rueda et al., 2013*; *Gomes et al., 2017*) and *P. costatus* (*Barroca et al., 2012a*, *2012b*).

This study tests the hypothesis that *P. magdalenae* exhibits genetic stocks that coexist and co-migrate throughout the rivers, tributaries, and floodplain lakes of the different Colombian Magdalena-Cauca and Caribbean hydrographic areas. Additionally, we compare the genetic diversity and structure with those of five sites (Pijiño, Llanito, Mompox, Palomino and San Marcos) previously studied by *Orozco-Berdugo & Narváez-Barandica (2014)*. To test this hypothesis, we developed species-specific microsatellite loci due to their advantages in the studies of population genetics (*Fernandez-Silva et al., 2013*; *Putman & Carbone, 2014*).

# MATERIALS AND METHODS

## Sample collection

This study analyzed a total of 725 muscle tissues of *P. magdalenae* from the river mainstream and floodplain lakes along the different Colombian Magdalena-Cauca and Caribbean hydrographic areas (Fig. 1; Supplemental File S1) that included 40 juveniles from a local fish hatchery. The samples preserved in 70% ethanol were provided by Integral S.A. through two scientific cooperation agreements (19 September 2013; Grant CT-2013-002443). Sampling collection was performed by Integral S. A., framed under an environmental permit from Ministerio de Ambiente, Vivienda y Desarrollo Territorial de Colombia # 0155 (30 January 2009) for Ituango hydropower construction. Samples previously studied by *Orozco-Berdugo & Narváez-Barandica (2014)* were collected during project 111752128352 of COLCIENCIAS under collection permit #1293 (2013) of the Universidad del Magdalena. The morphological diagnostic of individuals of *P. magdalenae* was conducted following the taxonomic keys of *Castro & Vari (2004)* and performed independently by both institutions.

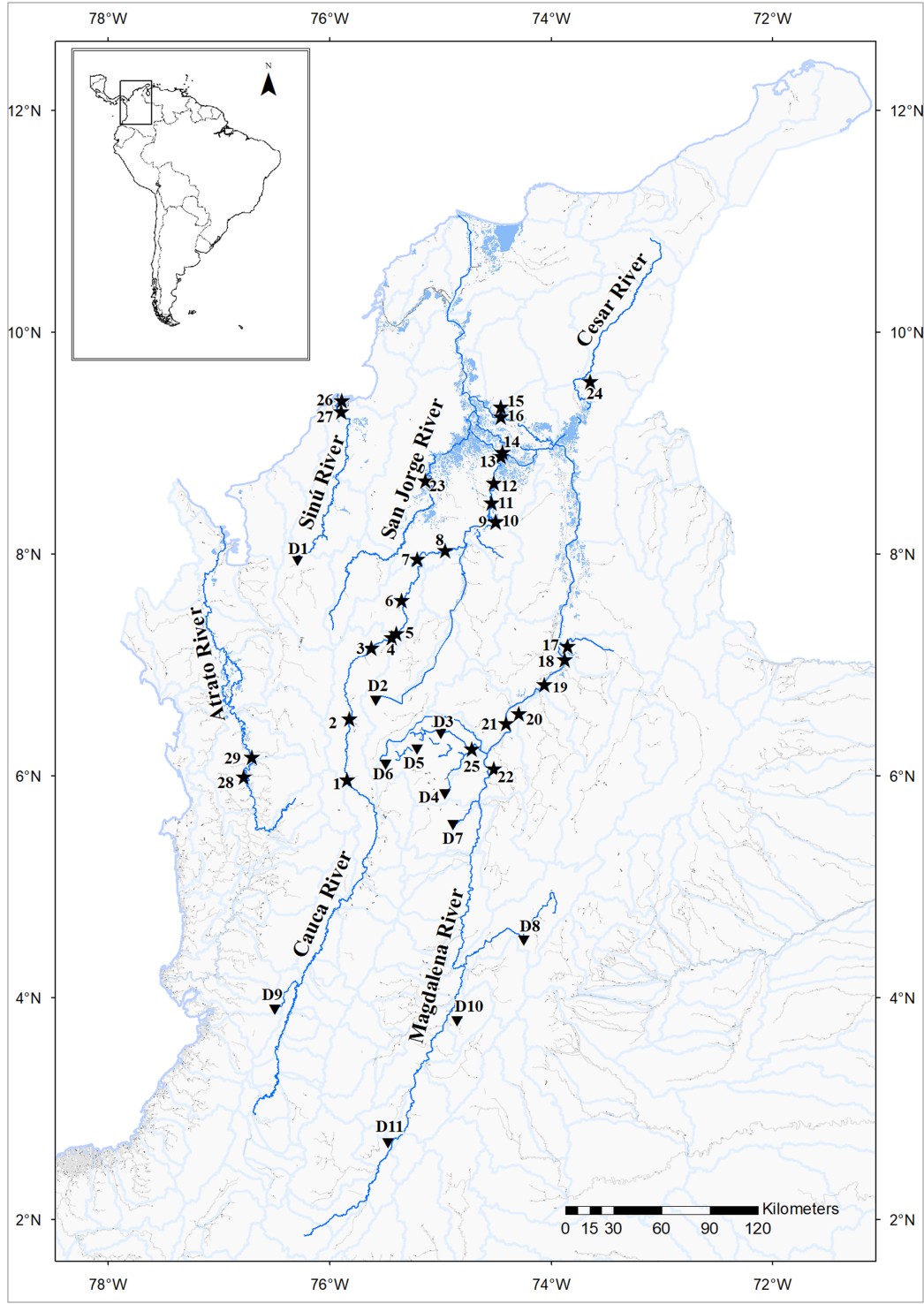

**Figure 1** *Prochilodus magdalenae* **sampling sites (stars) in the Colombian Magdalena-Cauca and Caribbean hydrographic areas.** Cauca River: Antioquia Department: Bolombolo (1); Puente Real (2); Gurimán (3); Espíritu Santo River (4); Puerto Valdivia (5); Cáceres (6); Man River (7); Margento (8). Bolívar Department: Floodplain Lakes Grande (9); Caimanera (10) and Panela (13); Achí (12). Sucre Department: Guaranda (11). Magdalena River: Bolívar Department: Palomino (14); Mompox (16). Magdalena Department: Pijiño Floodplain Lake (15). Santander Department: Barrancabermeja (18); Floodplain Lakes Llanito (17); Chucurí (19); Río Viejo (20). Antioquia Department: Puerto Berrío (21).

**Figure 1** (continued)
Boyacá Department: Palagua Floodplain Lake (22). San Jorge River: San Marcos River, Sucre Department (23). Cesar River: Cesar Department: Mata de Palma Floodplain Lake, El Paso (24). Nare River: Samaná Norte River, Antioquia Department (25). Sinú River: Córdoba Department: Caño Grande (26); Doctrina (27). Atrato River: Antioquia Department: Palo Blanco (29). Chocó Department: Beté (28). Dams (triangles): D1: Urra I; D2: Ríogrande; D3: San Lorenzo; D4: Playas; D5: El Peñol; D6: La Fe; D7: Miel; D8: Muña; D9: Calima; D10: Río Prado; D11: Betania. Map image layer by Instituto Geográfico Agustín Codazzi (IGAC).

## Microsatellite loci development

Low-coverage sequencing of the genomic library of one specimen of *P. magdalenae* from the middle section of the Magdalena River was performed using the Illumina MiSeq v2 platform (Illumina, CA, USA), the "whole genome shotgun" strategy and the Nextera library preparation kits (Illumina, CA, USA) for the sequence reads. All steps regarding the read cleaning, contig assemblage, identification of microsatellite loci, primer design, in silico alignment of primers using primer-BLAST (available in https://ncbiinsights.ncbi.nlm.nih.gov/2017/06/28/e-pcr-is-retiring-use-primer-blast/), PCR optimization, and polymorphism analysis of 52 microsatellites were performed following the methodology described by *Landínez-García & Márquez (2016)*. A set of 21 polymorphic microsatellite loci were selected and fluorescently labeled for genotyping of 88 randomly chosen samples. Then, a subset of 11 loci were selected for further evaluation of genetic diversity and structure in 725 samples because they satisfied the criteria of clearly defined peaks, reproducibility and consistency of amplifications, absence of stutter bands, specific bands, correct motif sizes, levels of heterozygosity, and high polymorphism information content (PIC) values, among others parameters required to validate new microsatellite primers (*Neff, Garner & Pitcher, 2011*; *Fernandez-Silva et al., 2013*; *Schoebel et al., 2013*).

## Genotyping of samples

The PCRs were conducted in a volume of 10 μl, which contained 2–4 ng/μl of template DNA isolated with the GeneJET Genomic DNA Purification kit (Thermo Scientific, Karlsruhe, Germany) following the manufacturer´s instructions, 1 × buffer (Invitrogen, CA, USA), 0.2 mM dNTPs (Thermo Scientific, MA, USA), 0.05 U/μl Platinum™ Taq DNA Polymerase (Invitrogen, CA, USA), 2.5 mM MgCl₂, 2% formamide (Sigma–Aldrich, Steinheim, Germany), 0.35 pmoles/μl labeled forward primer (either FAM6, VIC, NED, or PET, Applied Biosystems, CA, USA), and 0.5 pmoles/μl reverse primer (Macrogen, Seoul, Korea). The PCRs were performed on a T100 thermocycler (BioRad, CA, USA) with an initial denaturation step of 95 °C for 3 min followed by 32 cycles consisting of a denaturation step of 90 °C for 22 s and an annealing step for 18 s using the annealing temperatures described for each primer in Table 1. The extension step and a final elongation were absent in this thermal profile. Finally, the PCRs were submitted to electrophoresis on an automated sequencer ABI 3730 XL (Applied Biosystems, CA, USA) using GeneScan 500 LIZ dye size standard (Applied Biosystems, CA, USA) as the internal molecular size. Allelic fragments were denoted according to their molecular size and scored using GeneMapper v4.0 software (Applied Biosystems, CA,

**Table 1** Primer sequences, characteristics, polymorphism levels, and genetic diversity of 21 species-specific microsatellite loci in 88 individuals of *Prochilodus magdalenae* randomly chosen from the whole sample.

| Locus | Primer sequence (5′–3′) | Motif | Ta (°C) | Na | Ra | PIC | $H_O$ | $H_E$ | *P* |
|---|---|---|---|---|---|---|---|---|---|
| Pma39[a] | F: CCAATGACCTGTTTTCTACATTTGG | (ATCT)n | 58 | 14 | 231–283 | 0.860 | 0.671 | 0.878 | **0.002** |
| | R: AATCTACTACCCGGATGGCG | | | | | | | | |
| Pma25[a] | F: AAGGGGAAAGAAATCCAGGC | (AAGGC)n | 60 | 12 | 174–229 | 0.816 | 0.795 | 0.840 | **0.003** |
| | R: ATCCTGGGTTCATACCGACG | | | | | | | | |
| Pma02[a] | F: CGACATTCAACATGACAGTGC | (ATCT)n | 58 | 19 | 231–307 | 0.917 | 0.816 | 0.927 | **0.019** |
| | R: CACCAAATTGATGCAAACTGC | | | | | | | | |
| Pma35[a] | F: GCAGTCTGGCATTTTAGTGGC | (ATCT)n | 58 | 21 | 269–353 | 0.935 | 0.536 | 0.944 | **0.000** |
| | R: ACCACATCTCGCATCACTGG | | | | | | | | |
| Pma56[c] | F: ATTTGGTGCCTGTAGCTGGG | (ATT)n | 60 | 37 | 132–279 | 0.949 | 0.670 | 0.956 | **0.000** |
| | R: ACGGTCGGTGCACTAATTCC | | | | | | | | |
| Pma01[a] | F: TTGTCATTTCCCGGTTTTCC | (ATCT)n | 58 | 25 | 216–344 | 0.938 | 0.753 | 0.947 | **0.000** |
| | R: TGGCCCAGCTGTAATTTGG | | | | | | | | |
| Pma40[a] | F: CTGGTTACCCACCACTGTCG | (ATCT)n | 58 | 25 | 236–344 | 0.932 | 0.686 | 0.941 | **0.000** |
| | R: CACATTGCCATTTGGAGACG | | | | | | | | |
| Pma46[a] | F: TTGATGTAAACATCTCATTGCCG | (ATCT)n | 56 | 19 | 126–198 | 0.918 | 0.830 | 0.929 | **0.005** |
| | R: TTGCTGGAGGTTCTGTCCG | | | | | | | | |
| Pma36[a] | F: TCATGATGAAATGCCACACC | (ATCT)n | 58 | 24 | 119–219 | 0.925 | 0.674 | 0.935 | **0.000** |
| | R: TGCACGTGAACTTAGGCACC | | | | | | | | |
| Pma18[a] | F: ACTGAGACAAAACCCGGAGG | (ATT)n | 62 | 13 | 209–251 | 0.728 | 0.471 | 0.755 | **0.000** |
| | R: CTTCATACACCCACCATCAGG | | | | | | | | |
| Pma13[a] | F: CCGAAGCTATTTACCCAGCG | (AAAT)n | 62 | 11 | 154–194 | 0.815 | 0.670 | 0.841 | **0.007** |
| | R: TGAAATATGCTCGTGCTCCC | | | | | | | | |
| Pma14[a] | F: GTTCAGGGTCCTGCTGTTCC | (TTC)n | 58 | 21 | 146–209 | 0.907 | 0.605 | 0.919 | **0.000** |
| | R: TTTCGGTGTTGGAACATTGC | | | | | | | | |
| Pma42[c] | F: TTACACAGCGTCCCAATTCC | (ATCT)n | 58 | 25 | 146–254 | 0.933 | 0.759 | 0.942 | **0.000** |
| | R: GCTGCAGGGATTGTCCTACC | | | | | | | | |
| Pma26[c] | F: TGATGTTTCCTCCCCTCACC | (ATCTC)n | 58 | 20 | 141–281 | 0.888 | 0.553 | 0.902 | **0.000** |
| | R: GTGTTTCCTGCTCTCTGCCC | | | | | | | | |
| Pma34[d,e] | F: GAGTGCCGATGACAGAGACG | (ATCT)n | 58 | 24 | 202–406 | 0.919 | 0.363 | 0.930 | **0.000** |
| | R: CAAGATGCCCTGTAGTGCCC | | | | | | | | |
| Pma50[c] | F: GATTCCTTCCTACCGGAGCC | (ATCT)n | 58 | 30 | 171–299 | 0.942 | 0.565 | 0.950 | **0.000** |
| | R: ATGAGCACCACCCTCAATCC | | | | | | | | |
| Pma32[f] | F: GAAAAGACACAACAGCGCCC | (ATCT)n | 58 | 13 | 146–294 | 0.399 | 0.375 | 0.430 | **0.006** |
| | R: GTCGCTAATAGCCATGCCG | | | | | | | | |
| Pma57[b] | F: ATGGCAATGGTTAAGGGTCG | (AAC)n | 58 | 11 | 191–230 | 0.838 | 0.306 | 0.861 | **0.000** |
| | R: CTGAAAGCCCCTGTTTGTGC | | | | | | | | |
| Pma08[b,e] | F: TTTTATTATTCCCCATTTTCTCCC | (AAAG)n | 58 | 12 | 254–298 | 0.833 | 0.257 | 0.856 | **0.000** |
| | R: TGGGTTTTGAGCTGTTCTGC | | | | | | | | |
| Pma17[b] | F: CTGTGGGCAGCAAAGTGC | (ATT)n | 58 | 36 | 151–346 | 0.892 | 0.595 | 0.904 | **0.000** |
| | R: CTTTGAGCCACTTCAAACGG | | | | | | | | |

| Table 1 (continued) | | | | | | | | | |
|---|---|---|---|---|---|---|---|---|---|
| Locus | Primer sequence (5′–3′) | Motif | Ta (°C) | Na | Ra | PIC | $H_O$ | $H_E$ | *P* |
| Pma47[b] | F: TGGCTGCTAAATTAAATCCTTTGG | (ATCT)n | 58 | 21 | 176–280 | 0.915 | 0.413 | 0.928 | **0.000** |
| | R: AAGCAAAACCGTTCCACAGC | | | | | | | | |
| Across loci | | | | 20.619 | 119–353 | 0.867 | 0.589 | 0.876 | **0.000** |

Notes:
Ta: annealing temperature standardized in PCRs, Na: number alleles per locus; Ra: allelic size range (bp); PIC: polymorphism information content; $H_O$ and $H_E$: observed and expected heterozygosities, respectively; *P*: statistical significance (values in bold represent significance at *P* < 0.05).
[a] Satisfied selection criteria.
[b] inconsistent amplifications.
[c] Low definition peaks.
[d] Dropout.
[e] Stuttering.
[f] Low value of PIC.

USA; Supplemental File S2). Before the statistical analysis, Micro-Checker v.2.2.3 (*Van Oosterhout et al., 2004*) was run to detect potential genotyping errors.

## Statistical analysis

Tests for Hardy–Weinberg and Linkage equilibria, observed ($H_O$) and expected ($H_E$) heterozygosities and inbreeding coefficient ($F_{IS}$) were estimated using Arlequin v3.5.2.2 software (*Excoffier, Laval & Schneider, 2005*). The sequential Bonferroni correction was applied to adjust the statistical significance in multiple comparisons (*Holm, 1979*; *Rice, 1989*). The average number of alleles per locus and the PIC (*Botstein et al., 1980*) for each microsatellite locus were calculated with GenAlEx v6.503 software (*Peakall & Smouse, 2006*) and Cervus v3.0.7 software (*Marshall et al., 1998*), respectively.

The genetic differentiation among geographical samples was calculated using the standardized statistics $F'_{ST}$ (*Wright, 1943*, *1965*; *Meirmans, 2006*), Jost's $D'_{est}$ (*Jost, 2008*; *Meirmans & Hedrick, 2011*) and analysis of molecular variance (AMOVA) (*Meirmans, 2006*) with 10,000 permutations and bootstraps included in GenAlEx v6.503 software (*Peakall & Smouse, 2006*). Furthermore, the diploid genotypes of 11 loci (22 variables) in 725 individuals were submitted to discriminant analysis of principal components (DAPC) using the R-package Adegenet (*Jombart, 2008*).

To examine other groupings of the samples, genetic differentiation among samples was tested using the Bayesian analysis of population partitioning with Structure v2.3.4 software (*Pritchard, Stephens & Donnelly, 2000*). Parameters included 350,000 Monte Carlo Markov Chain steps and 50,000 iterations as burn-in, the admixture model, correlated frequencies, and the LOCPRIOR option for detecting relatively weak population structure (*Hubisz et al., 2009*). Each analysis was repeated 20 times for each simulated *K* value, which ranged from 1 to *n* + 3 (*n*, number of populations compared). For a best estimation of genetic stocks (K), the web-based software STRUCTURESELECTOR (*Li & Liu, 2018*) was used to calculate the Δ*K* ad hoc statistic (*Evanno, Regnaut & Goudet, 2005*), the estimators MEDMEANK, MAXMEANK, MEDMEDK and MAXMEDK (*Puechmaille, 2016*), and to generate the graphical representation of results using the integrated Clumpak software (*Kopelman et al., 2015*). Based on the coancestry coefficients provided by Structure and Clumpp, the individuals

were reorganized by genetic stock in sample sites that showed multiple stocks and were later submitted to the genetic analyses described above.

Additionally, the occurrence of recent genetic bottlenecks of populations was evaluated by calculating the levels of heterozygosity and the M ratio using Bottleneck v1.2.02 software v3.5.2.2 (*Piry, Luikart & Cornuet, 1999*) and Arlequin (*Excoffier, Laval & Schneider, 2005*), respectively. Excess heterozygosity was assessed by employing the Wilcoxon sign-rank test (*Cornuet & Luikart, 1996*; *Luikart & Cornuet, 1998*). The M ratio—the mean ratio of the number of alleles compared to the range of allele size— indicates that the population has experienced a recent and severe reduction in population size when its value smaller than 0.680 is (*Garza & Williamson, 2001*).

To explore non-neutral evolutionary forces acting on the microsatellite loci, a scanning analysis was performed using the BayeScan v2.1 software (*Foll & Gaggiotti, 2008*) to detect candidate loci under selection. Parameters for BayeScan analyses included 10:1 prior odds for the neutral model and 20 pilot runs consisting of 5,000 iterations each followed by 250,000 iterations with a burn-in length of 50,000 iterations (*Foll & Gaggiotti, 2008*).

## Phylogenetic relationships among genetic groups

To explore the phylogenetic relationships among individuals sampled along the basin, partial fragments of the mitochondrial *cox1* gene (~650 bp) were amplified in a subset of samples using primers and PCR conditions previously described by *Ward et al. (2005)* and *Ivanova et al. (2007)*. PCR products were sequenced by the Sanger method using an automated sequencer, ABI 3730 XL (Applied Biosystems, CA, USA). The best-fit evolutionary model was determined based on the Bayesian information criterion as implemented in the jModelTest v2.1.7 software (*Posada & Crandall, 1998*). A Bayesian phylogenetic analysis was conducted in MrBayes v3.2.6 software (*Ronquist & Huelsenbeck, 2003*) including GenBank sequences of *Prochilodus magdalenae*, *Prochilodus reticulatus*, *Prochilodus mariae*, *Prochilodus nigricans* and using *Ichthyoelephas longirostris* as outgroup. For this purpose, we performed two independent runs of 20 million generations sampled each 1,000 generations using 25% as burn-in. The remaining values were left as default. The convergence of each parameter was checked based on a potential scale reduction factor nearing 1, an average standard deviation of the split frequencies lower than 0.010, and the visualization of the resulting trees was performed with FigTree v1.4.3 software (*Rambaut, 2012*). Finally, the pair-wise divergences of *P. magdalenae* and *P. reticulatus* haplotype sequences were estimated using the Kimura 2-parameters model in MEGA v10.1.8 software (*Kumar et al., 2018*).

## RESULTS

### Microsatellite loci development

Genomic sequencing of the Illumina shotgun library of *P. magdalenae* (0.115 GB) generated 277,133 reads and 14,124 of 50,404 that contained microsatellite loci, were flanked by suitable PCR priming sites. The dinucleotides (47.758%) were the most abundant repeat motifs, followed by tetranucleotide (28.353%), trinucleotide (16.193%),

pentanucleotide (5.146%), and hexanucleotide (2.549%) repeats. The most common motifs found were AC (29.661%), TC (18.905%), ATT (4.811%) and AAAT (4.794%). The sequences of contigs containing the microsatellite loci obtained in the present study are provided in Supplemental Files S3 and S4.

A total of 21 of the 52 microsatellite loci evaluated were polymorphic and showed Hardy–Weinberg disequilibrium (Table 1) and Linkage equilibrium (Supplemental File S5). The number of alleles per locus ranged from 11 to 37, with an average number of 20.619 alleles/locus, the average values of observed and expected heterozygosities were $H_O = 0.589$ and $H_E = 0.876$, and the PIC values ranged from 0.399 to 0.949 (average 0.867) (Table 1). A total of 10 loci failed to satisfy the selection criteria, showing low values of PIC (Pma32), dropout and stuttering (Pma32, Pma08), inconsistent amplifications (Pma17, Pma47, Pma57), or low-definition peaks (Pma42, Pma56, Pma26, Pma50). Consequently, only 11 (Pma39, Pma25, Pma02, Pma35, Pma01, Pma40, Pma46, Pma36, Pma18, Pma13 and Pma14) satisfied most of the parameters required to validate the new microsatellite primers described previously.

## Genetic diversity, population demography and outlier loci screening

Comparisons among rivers revealed that 8 of 11 loci satisfied the Hardy–Weinberg equilibrium expectations in at least one case (Table 2). However, the analysis across loci showed significant departures from Hardy–Weinberg equilibrium expectations in all rivers evaluated (Table 2). The average number of alleles per locus was higher in Cauca (22.455) and Magdalena (19.455), followed by Nare (15.636), Sinú (15.273), the fish hatchery (14.818) and Atrato (14.636) and was lowest in San Jorge (13.545) and Cesar (13.364). Additionally, the highest values of observed and expected heterozygosities were found in San Jorge ($H_O$: 0.809; $H_E$: 0.884) and Cesar ($H_O$: 0.782; $H_E$: 0.873), followed by Sinú ($H_O$: 0.767; $H_E$: 0.882), Magdalena ($H_O$: 0.758; $H_E$: 0.896), and Cauca ($H_O$: 0.725; $H_E$: 0.898) and were lowest in Atrato ($H_O$: 0.718; $H_E$: 0.879), the fish hatchery ($H_O$: 0.691; $H_E$: 0.880), and Nare ($H_O$: 0.659; $H_E$: 0.876) (Table 2).

Furthermore, comparisons among sites within each river showed similar high levels of genetic diversity (Table 3). The highest value of genetic diversity was found in the floodplain lake Palagua in the Magdalena River (Na: 17.182 alleles/locus; $H_E$: 0.895; $H_O$: 0.792), whereas the lowest was observed in Beté, a site of the Atrato River (Na: 9.273 alleles/locus; $H_E$: 0.791; $H_O$: 0.711). In addition, all sites exhibited a highly significant deficit of observed heterozygosity (Table 3) with Mata de Palma and Samaná Norte River showing the lowest and highest observed heterozygosity deficits, respectively. Inbreeding coefficients ($F_{IS}$) per site in main rivers of the different Colombian hydrographic areas were significant and ranged from 0.120 to 0.255 (Table 3). Although decreased in magnitude, the inbreeding coefficients (Table 3) remained significant even after comparing the genetic diversity according to genetic stocks in Chucurí, Puerto Berrío, and Palagua and among the Magdalena River and tributaries.

Results of the genetic bottleneck tests (Table 4) were significant for all populations under the infinite alleles model (IAM) and for most populations under the two-phase model (TPM), whereas they were non-significant under the stepwise mutation

Table 2 Genetic diversity of *Prochilodus magdalenae* in main rivers of the distribution range of the species in Colombian hydrographic areas.

| River (N) | Diversity | Pma39 | Pma25 | Pma02 | Pma35 | Pma01 | Pma40 | Pma46 | Pma36 | Pma18 | Pma13 | Pma14 | Across loci |
|---|---|---|---|---|---|---|---|---|---|---|---|---|---|
| Cauca | Na | 19.000 | 15.000 | 25.000 | 25.000 | 34.000 | 28.000 | 21.000 | 25.000 | 17.000 | 13.000 | 25.000 | 22.455 |
| (308) | $H_O$ | 0.662 | 0.805 | 0.883 | 0.591 | 0.756 | 0.708 | 0.818 | 0.688 | 0.552 | 0.821 | 0.685 | 0.725 |
| | $H_E$ | 0.889 | 0.855 | 0.935 | 0.935 | 0.941 | 0.944 | 0.920 | 0.932 | 0.775 | 0.842 | 0.926 | 0.898 |
| | *P* | **0.000** | **0.000** | **0.002** | **0.000** | **0.000** | **0.000** | **0.001** | **0.000** | **0.000** | **0.000** | **0.000** | **0.000** |
| Magdalena | Na | 15.000 | 12.000 | 21.000 | 22.000 | 31.000 | 26.000 | 18.000 | 21.000 | 15.000 | 12.000 | 21.000 | 19.455 |
| (232) | $H_O$ | 0.664 | 0.891 | 0.861 | 0.642 | 0.781 | 0.679 | 0.818 | 0.745 | 0.599 | 0.854 | 0.803 | 0.758 |
| | $H_E$ | 0.874 | 0.865 | 0.930 | 0.941 | 0.943 | 0.944 | 0.925 | 0.926 | 0.784 | 0.833 | 0.929 | 0.896 |
| | *P* | **0.000** | **0.001** | 0.510 | **0.000** | **0.000** | **0.000** | **0.000** | **0.000** | **0.000** | 0.058 | **0.002** | **0.000** |
| San Jorge | Na | 10.000 | 11.000 | 16.000 | 19.000 | 16.000 | 18.000 | 14.000 | 14.000 | 9.000 | 9.000 | 13.000 | 13.545 |
| (20) | $H_O$ | 0.850 | 1.000 | 0.950 | 0.700 | 0.950 | 0.750 | 0.900 | 0.800 | 0.850 | 0.700 | 0.450 | 0.809 |
| | $H_E$ | 0.881 | 0.878 | 0.947 | 0.951 | 0.947 | 0.942 | 0.918 | 0.914 | 0.831 | 0.851 | 0.912 | 0.884 |
| | *P* | 0.650 | 0.299 | 0.645 | **0.000** | 0.638 | **0.002** | 0.531 | 0.307 | **0.009** | 0.318 | **0.000** | **0.000** |
| Cesar | Na | 10.000 | 9.000 | 15.000 | 16.000 | 21.000 | 15.000 | 13.000 | 17.000 | 9.000 | 8.000 | 14.000 | 13.364 |
| (20) | $H_O$ | 0.500 | 0.950 | 1.000 | 0.750 | 1.000 | 0.650 | 1.000 | 0.800 | 0.600 | 0.800 | 0.550 | 0.782 |
| | $H_E$ | 0.867 | 0.874 | 0.940 | 0.949 | 0.954 | 0.940 | 0.924 | 0.927 | 0.815 | 0.776 | 0.883 | 0.873 |
| | *P* | **0.000** | 0.890 | 0.947 | **0.033** | 0.208 | **0.002** | 0.484 | 0.148 | 0.097 | 0.846 | **0.000** | **0.000** |
| Nare | Na | 13.000 | 13.000 | 19.000 | 18.000 | 25.000 | 19.000 | 14.000 | 20.000 | 8.000 | 8.000 | 15.000 | 15.636 |
| (41) | $H_O$ | 0.610 | 0.780 | 0.902 | 0.415 | 0.780 | 0.439 | 0.927 | 0.805 | 0.341 | 0.756 | 0.488 | 0.659 |
| | $H_E$ | 0.887 | 0.877 | 0.931 | 0.930 | 0.952 | 0.931 | 0.912 | 0.934 | 0.708 | 0.781 | 0.912 | 0.876 |
| | *P* | **0.002** | 0.200 | 0.619 | **0.000** | **0.011** | **0.000** | 0.792 | **0.001** | **0.000** | 0.357 | **0.000** | **0.000** |
| Sinú | Na | 13.000 | 12.000 | 19.000 | 19.000 | 23.000 | 18.000 | 14.000 | 15.000 | 8.000 | 10.000 | 17.000 | 15.273 |
| (34) | $H_O$ | 0.441 | 0.912 | 0.912 | 0.647 | 0.647 | 0.824 | 0.824 | 0.882 | 0.735 | 0.824 | 0.794 | 0.767 |
| | $H_E$ | 0.916 | 0.867 | 0.939 | 0.919 | 0.936 | 0.906 | 0.884 | 0.921 | 0.827 | 0.823 | 0.904 | 0.882 |
| | *P* | **0.000** | 0.064 | 0.129 | **0.000** | **0.000** | **0.004** | 0.074 | **0.004** | 0.143 | 0.089 | **0.036** | **0.000** |
| Atrato | Na | 11.000 | 9.000 | 17.000 | 20.000 | 22.000 | 21.000 | 15.000 | 15.000 | 7.000 | 6.000 | 18.000 | 14.636 |
| (30) | $H_O$ | 0.300 | 0.933 | 0.900 | 0.600 | 0.700 | 0.700 | 0.900 | 0.933 | 0.667 | 0.500 | 0.767 | 0.718 |
| | $H_E$ | 0.817 | 0.849 | 0.933 | 0.945 | 0.946 | 0.946 | 0.912 | 0.920 | 0.849 | 0.788 | 0.933 | 0.879 |
| | *P* | **0.000** | 0.409 | 0.257 | **0.000** | **0.000** | **0.000** | 0.511 | 0.995 | **0.010** | **0.003** | **0.002** | **0.000** |
| Fish Hatchery | Na | 11.000 | 9.000 | 19.000 | 16.000 | 23.000 | 18.000 | 14.000 | 18.000 | 9.000 | 8.000 | 18.000 | 14.818 |
| (40) | $H_O$ | 0.750 | 0.750 | 0.925 | 0.500 | 0.800 | 0.625 | 0.725 | 0.625 | 0.600 | 0.675 | 0.625 | 0.691 |
| | $H_E$ | 0.887 | 0.825 | 0.940 | 0.925 | 0.943 | 0.919 | 0.927 | 0.922 | 0.795 | 0.799 | 0.920 | 0.880 |
| | *P* | **0.030** | **0.014** | 0.625 | **0.000** | **0.007** | **0.000** | **0.001** | **0.000** | **0.000** | 0.137 | **0.000** | **0.000** |

Note:
N, sample size; Na, number alleles per locus; $H_O$ and $H_E$, observed and expected heterozygosity, respectively; *P*: statistical significance for tests of departure from Hardy–Weinberg equilibrium. Values in bold represent significance at $P < 0.05$.

model (SMM). As it is thought that few loci follow the strict SMM (*Piry, Luikart & Cornuet, 1999*), the best estimation of expected heterozygosity at mutation-drift equilibrium is expected under a combination of IAM and TPM. Additionally, all values of the M ratio were substantially smaller than 0.680, indicating that all populations have experienced recent and severe reductions in population size (Table 4).

In contrast to other samples that did not show evidence of selection, BayeScan analysis revealed that 8 of 11 loci (Pma39, Pma25, Pma02, Pma35, Pma40, Pma36,

**Table 3 Genetic diversity and inbreeding coefficient of *Prochilodus magdalenae* per site and per genetic stock suggested by Structure in the main rivers of the distribution range of the species in Colombian hydrographic areas.**

| River | Sampling site (N) | Na | $H_O$ | $H_E$ | P | $F_{IS}$ | P |
|---|---|---|---|---|---|---|---|
| Cauca | S1 (33) | 15.273 | 0.667 | 0.878 | **0.000** | 0.255 | **0.000** |
| | S2[1] (30) | 15.727 | 0.773 | 0.885 | **0.000** | 0.143 | **0.000** |
| | S3 (28) | 14.182 | 0.740 | 0.886 | **0.000** | 0.182 | **0.000** |
| | S4 (38) | 14.818 | 0.732 | 0.885 | **0.000** | 0.186 | **0.000** |
| | S5[1] (40) | 15.636 | 0.700 | 0.885 | **0.000** | 0.221 | **0.000** |
| | S6a[2] (34) | 14.455 | 0.706 | 0.864 | **0.000** | 0.197 | **0.000** |
| | S6b[2] (26) | 14.364 | 0.752 | 0.881 | **0.000** | 0.165 | **0.000** |
| | S6c (34) | 15.364 | 0.719 | 0.879 | **0.000** | 0.196 | **0.000** |
| | S8[2] (45) | 15.909 | 0.743 | 0.887 | **0.000** | 0.173 | **0.000** |
| Magdalena | Pijiño[2] (19) | 12.273 | 0.780 | 0.865 | **0.000** | 0.125 | **0.000** |
| | Mompox[1] (19) | 13.091 | 0.770 | 0.882 | **0.000** | 0.154 | **0.000** |
| | Palomino[1] (20) | 13.182 | 0.759 | 0.869 | **0.000** | 0.152 | **0.000** |
| | Río Viejo[2] (24) | 13.909 | 0.739 | 0.883 | **0.000** | 0.184 | **0.000** |
| | Llanito[2] (31) | 15.000 | 0.774 | 0.879 | **0.000** | 0.135 | **0.000** |
| | Barrancabermeja[1] (24) | 13.636 | 0.727 | 0.872 | **0.000** | 0.186 | **0.000** |
| | Chucurí (Ch)[2] (32) | 15.000 | 0.699 | 0.882 | **0.000** | 0.223 | **0.000** |
| | Puerto Berrío (B)[1] (28) | 14.818 | 0.714 | 0.883 | **0.000** | 0.208 | **0.000** |
| | Palagua (P)[2] (35) | 17.182 | 0.792 | 0.895 | **0.000** | 0.129 | **0.000** |
| | ChBP Stock1 (28) | 13.000 | 0.698 | 0.851 | **0.000** | 0.198 | **0.000** |
| | ChBP Stock2 (48) | 18.636 | 0.759 | 0.895 | **0.000** | 0.162 | **0.000** |
| | ChBP Stock3 (14) | 9.909 | 0.695 | 0.833 | **0.000** | 0.202 | **0.000** |
| Cauca + Magdalena-(ChBP) | Stock1 (241) | 21.182 | 0.723 | 0.893 | **0.000** | 0.192 | **0.000** |
| | Stock2 (285) | 21.727 | 0.742 | 0.895 | **0.000** | 0.172 | **0.000** |
| San Jorge | San Marcos River | 13.364 | 0.782 | 0.873 | **0.000** | 0.130 | **0.000** |
| Cesar | Mata de Palma | 13.545 | 0.809 | 0.884 | **0.000** | 0.110 | **0.000** |
| Nare | Samaná Norte River | 15.630 | 0.659 | 0.876 | **0.000** | 0.260 | **0.000** |
| Sinú | Caño Grande[1] (16) | 11.000 | 0.744 | 0.845 | 0.000 | 0.151 | 0.000 |
| | Doctrina[1] (18) | 11.545 | 0.788 | 0.867 | 0.000 | 0.120 | 0.001 |
| Atrato | Palo Blanco[1] (19) | 12.727 | 0.722 | 0.869 | **0.000** | 0.195 | **0.000** |
| | Beté[1] (11) | 9.273 | 0.711 | 0.791 | **0.000** | 0.149 | **0.000** |

**Notes:**
[1] Sampling site on the main stream.
[2] Sampling site on floodplain lakes. S6a: Floodplain Lake Grande, S6b: Floodplain Lake Caimanera F, S6c: Guaranda. N, sample size; Na, average number of alleles per locus; $H_O$ and $H_E$, observed and expected heterozygosity, respectively; $F_{IS}$, inbreeding coefficient; P, statistical significance for tests of departure from Hardy–Weinberg equilibrium. Values in bold represent significance at $P < 0.05$.

Pma13 and Pma14) exhibit substantial evidence of selection in the Magdalena River (Table 5).

## Genetic structure and phylogenetic relationships among the samples studied

At regional scale, the Bayesian analysis showed the presence of two genetic stocks ($\Delta K = 2$; MEDMEDK = 2; MEDMEANK = 2), one in the Magdalena River (Chucurí + Puerto

**Table 4 Tests to detect recent genetic bottleneck in *Prochilodus magdalenae* populations.**

| River/stock | IAM | SMM | TPM | M ratio value |
|---|---|---|---|---|
| Cauca (C) | **0.000** | 0.958 | **0.027** | 0.254 ± 0.037 |
| Magdalena (M) | **0.000** | 0.517 | **0.008** | 0.219 ± 0.032 |
| Sinú | **0.000** | 0.183 | **0.000** | 0.155 ± 0.026 |
| Atrato | **0.000** | 0.584 | 0.062 | 0.151 ± 0.022 |
| Fish Hatchery | **0.000** | 0.382 | **0.001** | 0.173 ± 0.022 |
| Chucurí (Ch) | **0.000** | 0.232 | **0.001** | 0.156 ± 0.067 |
| Puerto Berrío (B) | **0.000** | 0.074 | **0.000** | 0.154 ± 0.067 |
| Palagua (P) | **0.000** | 0.740 | **0.005** | 0.175 ± 0.051 |
| ChBP Stock1 | **0.000** | 0.958 | 0.103 | 0.160 ± 0.239 |
| ChBP Stock2 | **0.000** | 0.551 | **0.000** | 0.228 ± 0.050 |
| ChBP Stock3 | **0.002** | 0.551 | 0.160 | 0.126 ± 0.021 |
| CM Stock1 | **0.000** | 0.997 | **0.027** | 0.240 ± 0.044 |
| CM Stock2 | **0.000** | 0.966 | **0.003** | 0.245 ± 0.025 |

Note:
IAM, infinite alleles model; SMM, stepwise mutation model; TPM, two-phase model. Wilcoxon test probability (one tail for H excess) (*Luikart & Cornuet, 1998*) calculated by Bottleneck v1.2.02 (*Piry, Luikart & Cornuet, 1999*). M ratio value (*Garza & Williamson, 2001*), calculated by Arlequin v3.5.2.2 (*Excoffier, Laval & Schneider, 2005*).

**Table 5 Parameters estimated using Bayesian likelihood method for searching candidate loci under selection.**

| Locus | $P$ | $Log_{10}(PO)$ | $q$-Value | Alpha | $F_{ST}$ |
|---|---|---|---|---|---|
| Pma39 | 0.883 | 0.880 | 0.017 | −1.470 | 0.008 |
| Pma25 | 0.987 | 1.890 | 0.002 | −2.062 | 0.004 |
| Pma02 | 0.999 | 3.220 | 0.000 | −2.002 | 0.004 |
| Pma35 | 0.998 | 2.660 | 0.000 | −1.862 | 0.005 |
| Pma01 | 0.122 | −0.860 | 0.141 | 0.078 | 0.028 |
| Pma40 | 1.000 | 1000 | 0.000 | 1.210 | 0.082 |
| Pma46 | 0.048 | −1.300 | 0.215 | 0.000 | 0.026 |
| Pma36 | 1.000 | 1000 | 0.000 | −2.589 | 0.002 |
| Pma18 | 0.599 | 0.170 | 0.059 | 0.416 | 0.039 |
| Pma13 | 1.000 | 1000 | 0.000 | 1.384 | 0.095 |
| Pma14 | 1.000 | 1000 | 0.000 | −2.116 | 0.004 |

Note:
$P$: posterior probability of the model including selection; $Log_{10}(PO)$: the logarithm of posterior odds to base 10 for the model including selection; $q$-value: minimum false discovery rate at which a locus may become significant; Alpha: locus-specific component shared by all populations using a logistic regression, indicating the strength and direction of the selection; $F_{ST}$: coefficient to measure the difference in allele frequency between the common gene pool and each subpopulation, calculated as a posterior mean using model averaging.

Berrio + Palagua) and the other one in the remaining evaluated rivers (Fig. 2A), which is concordant with DAPC (Fig. 2B) and AMOVA ($F_{ST(7, 1407)} = 0.009$; $P = $ **0.000**). Together with Chucurí + Puerto Berrio + Palagua, a predominant genetic stock with different levels of genetic admixture in Sinú and Atrato rivers was revealed in the clusters suggested by the MAXMEANK and MAXMEDK statistics ($K = 3$). The additional clustering patterns ($K = 4$–8; Fig. 2A) examined to compared them with other approaches, showed genetic

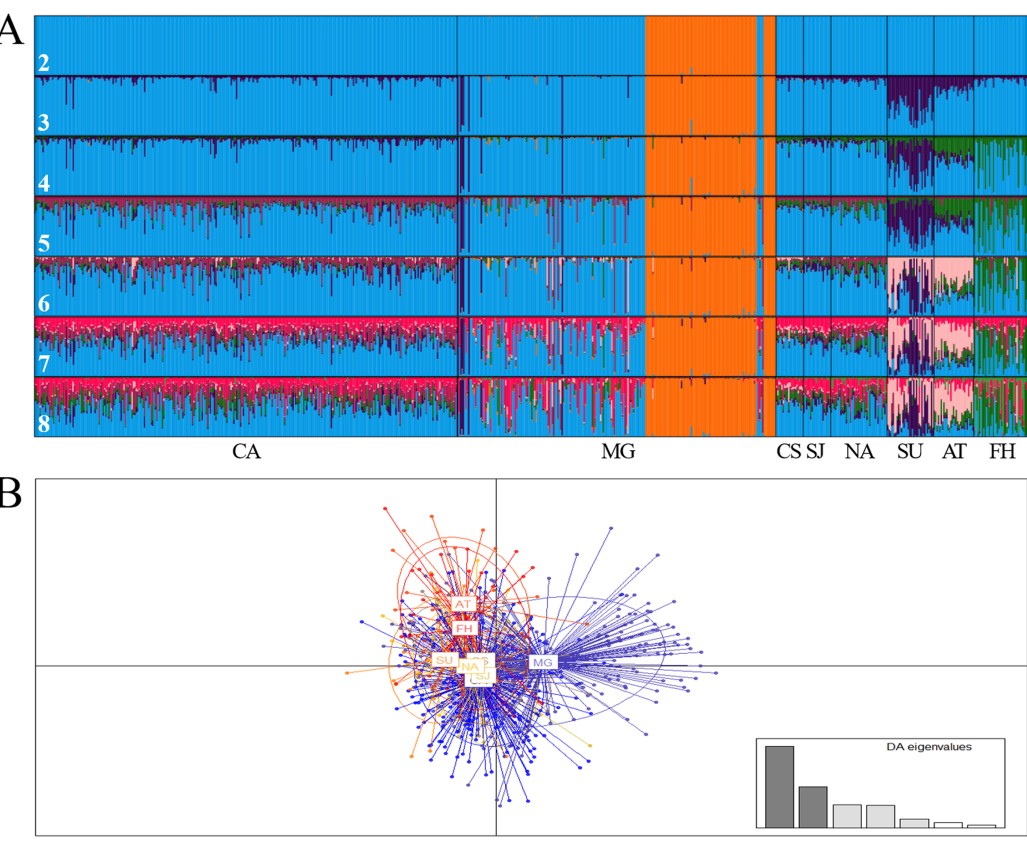

**Figure 2  Bar plot of population ancestry coefficients as estimated by Structure (A) and discriminant analysis of principal components (B) of *Prochilodus magdalenae* from the Colombian Magdalena-Cauca and Caribbean hydrographic areas.** CA: Cauca River; MG: Magdalena River; CS: Cesar River, SJ: San Jorge River; NA: Nare River; SU: Sinú River; AT: Atrato River; FH: fish hatchery. The numbers denote estimated genetic stocks (K).

admixture with other stocks absent in the examined rivers. Likewise, pairwise comparisons of the standardized statistics $F'_{ST}$ (*Meirmans, 2006*) and Jost's $D'_{est}$ (*Meirmans & Hedrick, 2011*) showed genetic differences among Atrato, the fish hatchery, Sinú, and the remaining rivers (Table 6) as well as among the Magdalena River and its tributaries, Cauca and Nare.

However, excluding samples that exhibit loci putatively under selection (Chucurí + Puerto Berrio + Palagua), comparisons among sites within each river revealed a genetic admixture of two stocks ($\Delta K = 2$; MEDMEDK = 2; MEDMEANK = 2) homogenously distributed in Magdalena River and its tributaries (Figs. 3A and 3B; Tables 6 and 7). Additionally, this analysis revealed a genetic substructure in Sinú ($\Delta K = 2$; MEDMEDK = 2; MEDMEANK = 2; Figs. 3C and 3D; $F_{ST(1, 67)} = 0.033$; $P = \textbf{0.000}$; $F'_{ST} = 0.027$; $P = \textbf{0.004}$; $D'_{est} = 0.149$; $P = \textbf{0.005}$). In Atrato River, the Bayesian analysis showed a single genetic stock ($\Delta K = 2$; MEDMEDK = 1; MEDMEANK = 1; Fig. 3E) although remaining analysis showed genetic differentiation among sites (Fig. 3F; $F_{ST(1, 57)} = 0.045$; $P = \textbf{0.000}$; $F'_{ST} = 0.047$; $P = \textbf{0.000}$; $D'_{est} = 0.330$; $P = \textbf{0.000}$).

**Table 6 Pairwise Jost's $D_{est}$ (upper diagonal) and $F'_{ST}$ (below diagonal) of *Prochilodus magdalenae* samples among rivers of the distribution range of the species in Colombian hydrographic areas.**

| River/deme | 1 | 2 | 3 | 4 | 5 | 6 | 7 | 8 | 9 |
|---|---|---|---|---|---|---|---|---|---|
| 1. Cauca | | **0.065** | 0.009 | 0.010 | −0.003 | 0.020 | **0.146** | **0.146** | **0.105** |
| 2. Magdalena | **0.004** | | **0.033** | 0.052 | 0.047 | **0.086** | **0.219** | **0.182** | **0.134** |
| 3. Magdalena-ChBP | 0.002 | **0.003** | | 0.019 | −0.007 | 0.013 | **0.152** | **0.134** | **0.103** |
| 4. Cesar | 0.008 | 0.010 | 0.009 | | −0.010 | 0.025 | **0.104** | **0.139** | 0.042 |
| 5. San Jorge | 0.008 | 0.010 | 0.008 | 0.014 | | 0.007 | **0.108** | **0.156** | **0.114** |
| 6. Nare | 0.005 | **0.009** | 0.006 | 0.013 | 0.012 | | **0.156** | **0.132** | **0.097** |
| 7. Sinú | **0.013** | **0.016** | **0.014** | **0.017** | **0.018** | **0.017** | | **0.202** | **0.209** |
| 8. Atrato | **0.014** | **0.015** | **0.014** | **0.020** | **0.021** | **0.017** | **0.021** | | **0.149** |
| 9. Fish Hatchery | **0.010** | **0.011** | **0.011** | 0.013 | **0.018** | **0.014** | **0.020** | **0.018** | |

**Note:**
Values in bold denote statistical significance after Bonferroni correction ($P < 0.002$).

Finally, the Bayesian tree using the *cox1* gene clustered our samples (GenBank accession numbers MK330430–MK330494) with sequences of *P. magdalenae* and *P. reticulatus* deposited in public databases and in a different group, *P. mariae* and *P. nigricans* (Fig. 4). Moreover, Kimura-2-parameters genetic distances (Supplemental File S6) were larger among haplotypes of *P. magdalenae* (0.002–0.010) than among *P. magdalenae* and *P. reticulatus* haplotypes (0.000–0.005).

# DISCUSSION

## Microsatellite loci development

This work developed species-specific microsatellite loci using next-generation sequencing and bioinformatic analysis. Although a total of 21 of 52 microsatellite loci with tri- and tetra-nucleotide motifs were polymorphic in *P. magdalenae*, the consistency in the amplification in a larger sample, allelic size class distribution, and high definition peaks allowed the selection of only 11 microsatellite loci for further population genetic analysis. Most of the loci showed departures from Hardy–Weinberg equilibrium and significant observed heterozygosity deficit in the random sample. The observed heterozygosity deficit may be related to technical problems such as silent alleles; however, it remains to explore the potential variations in the primer alignment sequences, since this study sequenced the genome of a single specimen of *P. magdalenae*. Two non-excluding explanations may be related to the significant levels of inbreeding and the genetic structure of the samples by the coexistence of two genetic stocks (see below).

Although the levels of genetic diversity measured by the expected heterozygosities were similar, the levels of observed heterozygosity as well as the average number of alleles per locus found in this study were substantially greater than those found in the same samples by *Orozco-Berdugo & Narváez-Barandica (2014)*. However, despite these differences, both heterologous (*Orozco-Berdugo & Narváez-Barandica, 2014*) and species-specific microsatellite loci (this study) revealed a general deficit of heterozygotes in all samples. In this context, the species-specific microsatellite loci developed in this study

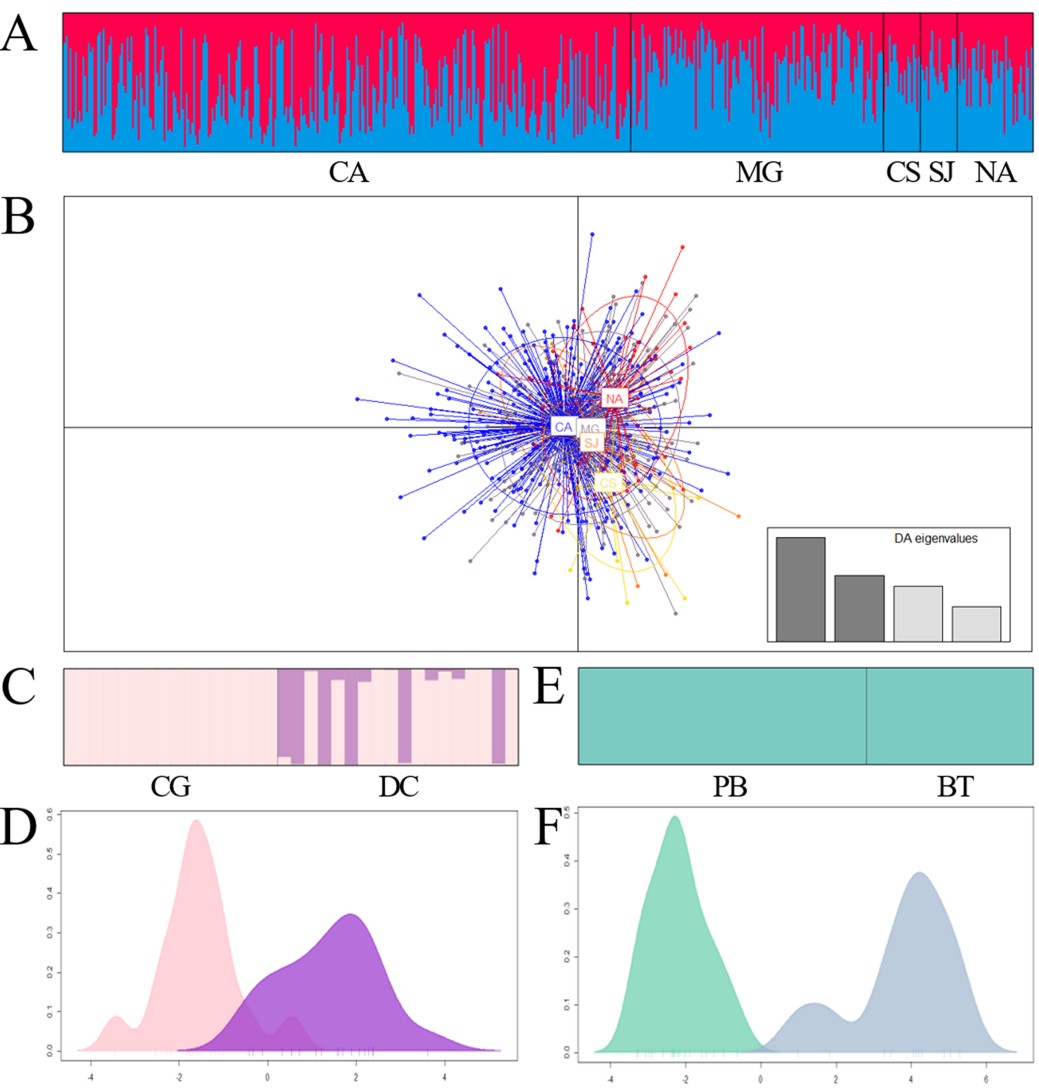

**Figure 3 Bar plot of population ancestry coefficients as estimated by Structure and discriminant analysis of principal components of *Prochilodus magdalenae* in the Magdalena River and tributaries (A and B), Sinú River (C and D) and Atrato River (E and F).** CA: Cauca River; MG: Magdalena River; CS: Cesar River; SJ: San Jorge River; NA: Nare River. CG: Caño Grande; DC: Doctrina; PB: Palo Blanco; BT: Beté.

seem to provide a good approach to study the population genetics of *P. magdalenae* considering that the levels of heterozygosity constitute a parameter used to estimate the genetic diversity of the populations. In addition to the applications in harvest management, stocking programs, definition of conservation units, recovery of threatened species, and management of invasive species, these tools may be useful in forensic genetics since partially degraded DNA samples are often found in this area (see *Bourret et al., 2020*).

## Genetic diversity and population demography

Microsatellite data revealed average values of genetic diversity ($H_E$: 0.737) among the highest values found in other Prochilodontidae species, only surpassed by those reported

**Table 7 Pairwise Jost's $D_{est}$ (upper diagonal) and $F'_{ST}$ (below diagonal) of *Prochilodus magdalenae* samples among sites of the rivers Cauca and Magdalena.**

| River | Sampling site | 1 | 2 | 3 | 4 | 5 | 6 | 7 | 8 | 9 |
|---|---|---|---|---|---|---|---|---|---|---|
| Cauca | 1. S1 | | 0.023 | 0.069 | 0.050 | 0.000 | 0.066 | 0.014 | 0.056 | 0.023 |
| | 2. S2 | 0.011 | | 0.059 | 0.006 | 0.007 | 0.020 | 0.020 | 0.036 | −0.003 |
| | 3. S3 | 0.014 | 0.013 | | 0.023 | 0.018 | **0.096** | 0.043 | 0.060 | 0.001 |
| | 4. S4 | 0.012 | 0.009 | 0.010 | | 0.007 | 0.062 | 0.045 | 0.056 | 0.018 |
| | 5. S5 | 0.009 | 0.009 | 0.010 | 0.008 | | 0.050 | 0.013 | 0.021 | 0.015 |
| | 6. S6a | 0.013 | 0.010 | **0.016** | 0.012 | 0.011 | | 0.052 | 0.038 | 0.044 |
| | 7. S6b | 0.011 | 0.012 | 0.013 | 0.012 | 0.010 | 0.013 | | 0.073 | 0.002 |
| | 8. S6c | 0.013 | 0.011 | 0.013 | 0.012 | 0.010 | 0.011 | 0.014 | | 0.003 |
| | 9. S8 | 0.009 | 0.008 | 0.009 | 0.008 | 0.008 | 0.010 | 0.009 | 0.008 | |
| Magdalena | 1. Pijiño | | 0.046 | 0.081 | 0.092 | 0.039 | 0.038 | **0.414** | **0.387** | **0.312** |
| | 2. Mompox | 0.018 | | 0.014 | 0.027 | −0.001 | 0.006 | **0.325** | **0.358** | **0.216** |
| | 3. Palomino | 0.020 | 0.016 | | 0.082 | 0.006 | −0.019 | **0.416** | **0.373** | **0.273** |
| | 4. Rio Viejo | 0.019 | 0.015 | 0.018 | | −0.005 | 0.013 | **0.400** | **0.411** | **0.277** |
| | 5. Llanito | 0.014 | 0.012 | 0.012 | 0.011 | | −0.041 | **0.381** | **0.395** | **0.245** |
| | 6. Barrancabermeja | 0.016 | 0.014 | 0.012 | 0.013 | 0.008 | | **0.356** | **0.350** | **0.238** |
| | 7. Chucurí | **0.036** | **0.029** | **0.035** | **0.032** | **0.031** | **0.031** | | 0.018 | 0.059 |
| | 8. Puerto Berrío | **0.035** | **0.031** | **0.033** | **0.033** | **0.032** | **0.031** | 0.011 | | −0.006 |
| | 9. Palagua | **0.028** | **0.022** | **0.026** | **0.024** | **0.022** | **0.023** | 0.012 | 0.009 | |

Note:
S6a: Floodplain Lake Grande, S6b: Floodplain Lake Caimanera F, S6c: Guaranda. Values in bold denote statistical significance after Bonferroni correction (Cauca: $P < 0.0005$; Magdalena: $P < 0.0001$).

for *P. costatus* (*Melo et al., 2013*) and *P. argenteus* (*Coimbra et al., 2017*) (0.747 and 0.753 respectively). Similarly, the average levels of expected heterozygosity were higher than that found in *P. magdalenae* measured by heterologous microsatellites ($H_E$: 0.877; *Orozco-Berdugo & Narváez-Barandica, 2014*) and Neotropical Characiforms ($H_E$: 0.675 ± 0.160; see review by *Hilsdorf & Hallerman (2017)*).

Additionally, this study found levels of observed heterozygosity higher than those found by *Orozco-Berdugo & Narváez-Barandica (2014)*. However, the use of species-specific microsatellite loci developed in this study revealed similar values of expected heterozygosity among samples analyzed by *Orozco-Berdugo & Narváez-Barandica (2014)* and the remaining samples analyzed, indicating that differences between the two studies are related to the type of microsatellite loci utilized (heterologous vs. species-specific microsatellite loci).

The significant deficit of observed heterozygosity in all studied samples corroborates the previous findings for *P. magdalenae* from Magdalena River (*Orozco-Berdugo & Narváez-Barandica, 2014*); however, the magnitude of the observed heterozygosity deficit as well as the inbreeding coefficient (0.075–0.239) were substantially lower than those previously reported (0.624–0.788). Following *Franklin (1980)* and *Soulé (1980)*, the values above 10% of the inbreeding coefficient indicate that these populations require careful management to avoid future detrimental effects on its populations. This point is important

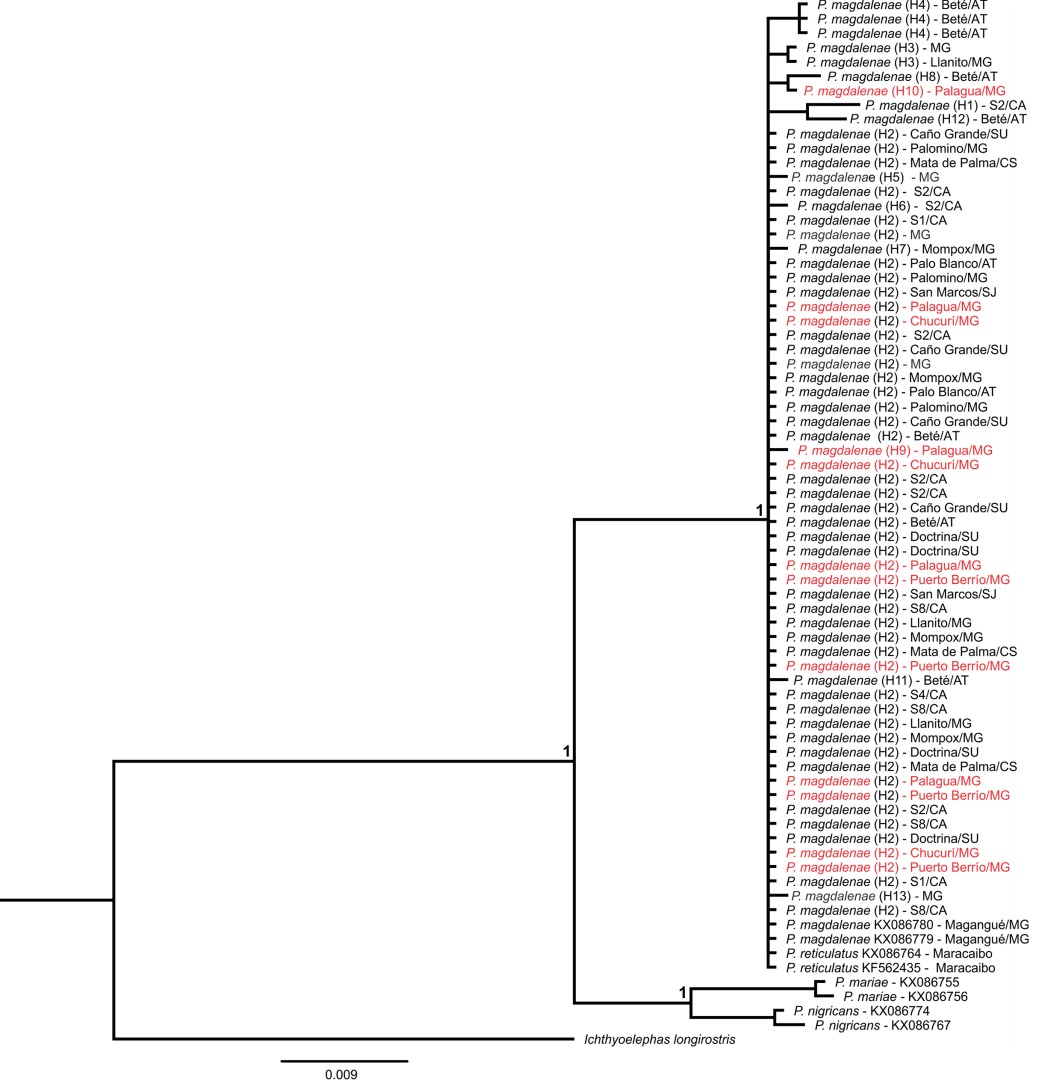

**Figure 4 Bayesian phylogenetic tree of *Prochilodus* based on partial sequences of *cox1* gene.** Node supports indicate posterior probability > 0.950. Red denote sequences from populations that exhibit outlier loci (Chucurí, Puerto Berrío, Palagua). Haplotypes are in parentheses. AT: Atrato River, CA: Cauca River; CS: Cesar River; MG: Magdalena River; SJ: San Jorge River; SU: Sinú River.

since it has been recommended recently that any inbreeding coefficient higher than zero will usually have an adverse fitness effect (*Frankham, Bradshaw & Brook, 2014*).

Another non-excluding alternative is plausible considering that the significant deficit of observed heterozygosity observed in all sites analyzed may be also explained by the coexistence of genetic stocks (Wahlund effect) as this was evidenced by the genetic structure analysis (see below). Another biological cause of observed heterozygosity deficit, assortative mating, does not seem to explain the results found in this study because *P. magdalenae* is iteroparous and characterized by total spawning (*Jaramillo-Villa & Jiménez-Segura, 2008*) as described in its congeners, *P. costatus* (*Carolsfield et al., 2004*) and

*P. lineatus* (*Roux et al., 2015*). Even more, in this latter species, the genetic analysis based on microsatellite loci support polygamous mating in both sexes (*Ribolli et al., 2020*).

On the other hand, this study also provided evidence for a population bottleneck, suggesting that *P. magdalenae* shows signs of erosion of the genetic pool, likely by the constant pressure from fishing and other anthropogenic activities exerted on its populations. Although paradoxical to the observed heterozygosity deficit found in all populations evaluated, this outcome is plausible considering that the Bottleneck algorithm tests not for an excess of heterozygotes ($H_O > H_E$) but rather for an excess of heterozygosity (He > He at mutation-drift equilibrium) (*Piry, Luikart & Cornuet, 1999*). Besides, the combination of a population bottleneck and an observed heterozygosity deficit may result from population growth in a closed system, population genetic structure, or admixture (*Barson, Cable & Oosterhout, 2009*). Considering the lengths of the rivers studied, population growth in a closed system is unlikely but the last two alternatives may explain our results due to the coexistence of genetic stocks in the samples studied and the continuous restocking of natural stocks using juveniles from fish hatcheries, which may create an apparent excess of novel alleles and an incomplete allele frequency distribution. Similar results have also been found in guppies, *Poecilia reticulata*, in Trinidad and Tobago (*Barson, Cable & Oosterhout, 2009*).

## Genetic structure

This study tested the hypothesis that *P. magdalenae* exhibits genetic stocks that coexist and co-migrate along sections of the main channel and some tributaries of the Magdalena River (Cauca, San Jorge and Cesar), Sinú, and Atrato rivers. Before testing this hypothesis, we compared the genetic structure at regional scale, finding two spatially structured populations: one in the Magdalena River (Puerto Berrío and the floodplains Chucurí and Palagua) and the other in the remaining rivers evaluated.

The geographical genetic structure may result from taxonomic differences among stocks due to the lack of regulations on the restocking of natural stocks of *P. magdalenae*. The phylogenetic analysis using partial sequences of *cox1* gene indicates that samples do not correspond to species such as *P. mariae* or *P. nigricans* because this genetic stock is clustered with previously published sequences of *P. magdalenae* (*Aguirre-Pabón, Narváez-Barandica & Castro-García, 2013*). However, it remains to be seen whether they represent artificial mixtures of *P. magdalenae* and *P. reticulatus* because the current mitochondrial phylogenetic analysis of Prochilodontidae does not allow the two species to be discriminated (*Melo et al., 2016b, 2018*). Moreover, the morphological and molecular similitudes have led to the proposal that *P. magdalenae* and *P. reticulatus* represent only one species with probable allopatric differentiation resulting from the uplift of the Sierra del Perijá (*Melo et al., 2016b*). Thus, a separated clustering of mitochondrial sequences of those stocks is not expected in the phylogenetic analysis even though they represent allopatric populations.

An alternative explanation is that the genetic differences result from eight outlier loci that are putatively under selection in three sites of the Magdalena River, suggesting that *P. magdalenae* experiences natural/artificial selection or local adaptation, although

testing of these hypotheses is out of the scope of the present study. The explanation that outlier loci represent false positives resulting from the inclusion of severely bottlenecked populations (*Teshima, Coop & Przeworski, 2006*; *Foll & Gaggiotti, 2008*) seems unlikely because the significant excess of heterozygosity and small values of the M ratio were found even in populations that do not exhibit outlier loci. Thus, considering that those sites have been exposed to restocking since 20 years ago and since most microsatellite loci are not transcriptionally active, the outlier loci found in this study may reflect hitchhiking selection resulting from restocking using juveniles selected artificially by fish hatcheries. Alternatively, the outlier loci may result from asymmetric gene flow by unidirectional migration from hatchery stocks to wild populations. Similar results were found in Denmark in populations of three brown trout, which have been significantly admixtured with stocked hatchery trout (*Hansen, Meier & Mensberg, 2010*).

Although the above reasoning might explain the genetic differences between stocks, an additional justification is required to explain the restricted distribution of one genetic stock in only three sites of the Magdalena River considering the migratory abilities of these species/allopatric populations. Thus, this genetic structure seems to result from recent restocking before reproductive/feeding migrations, use of artificial barriers to avoid migration of the fish, clogging by sedimentation or vegetation, or the desiccation of access to floodplain lakes or may be a product of the intensive anthropic intervention in these territories characterized by the exploitation of hydrocarbons and livestock. This idea is concordant with the fact that degradation of preferred habitat and barriers that impede dispersal contribute to the degree of genetic differentiation among populations (*Faulks, Gilligan & Beheregaray, 2011*).

Furthermore, the results found here provide support for the hypothesis that *P. magdalenae* exhibits genetic stocks that coexist and co-migrate along sections of the rivers Magdalena, Cauca, Cesar (tributaries of the Magdalena River), Sinú, and Atrato. Since similar patterns of genetic structure are found in *P. reticulatus* (*López-Macías et al., 2009*), *P. marggravii* (*Hatanaka & Galetti, 2003*), *P. argenteus* (*Sanches et al., 2012*), *P. costatus* (*Barroca et al., 2012a*), *P. magdalenae* (*Orozco-Berdugo & Narváez-Barandica, 2014*; *Hernández, Navarro & Muñoz, 2017*) and *Ichthyoelephas longirostris* (*Landínez-García & Márquez, 2016*), this outcome supports the idea that this genetic structure is a generalized tendency within the family Prochilodontidae.

Excluding the genetic stock of Puerto Berrío and the floodplains Chucurí and Palagua, each river showed the coexistence of at least two genetic stocks. Homogeneous and non-homogeneous distributions of these genetic stocks along the rivers explain similarities (Cauca, Magdalena, San Jorge, Cesar and Nare) as well as geographical differences among the rivers analyzed (within Magdalena, including Puerto Berrío and the floodplains Chucurí and Palagua, Sinú and Atrato). This genetic structure also explains the significant heterozygosity deficit observed in all sites analyzed (Wahlund effect) as discussed above. Similar evidence of the Wahlund effect has been documented in the congener *P. costatus*, which exhibited genetic differences resulting from temporal isolation (*Braga-Silva & Galetti, 2016*). Although sampling in this study was not designed to detect temporal genetic structuring, genetic similarities among samples collected in different

years suggest that the Wahlund effect must be more spatial than temporal. It remains to be seen whether this behavior is natural or artificial, considering that the restocking activities have been widely implemented along different Colombian rivers.

## CONCLUSIONS

This study provides evidence that *P. magdalenae* exhibits high genetic diversity, significant inbreeding levels per genetic stock, and signs of erosion of the genetic pool and conforms a mixture of genetic stocks heterogeneously distributed along the rivers studied. Additionally, this study developed a set of 11 microsatellite loci that allows the reliable detection of levels of genetic diversity, providing a tool for monitoring changes in the genetic diversity of the species, brood stocks and juveniles used for supportive breeding and for measuring the efficacy of current population restocking activities. Management and conservation strategies need to be implemented at the level of the basins Magdalena-Cauca, Sinú and Atrato concordantly with their genetic population structure.

## ACKNOWLEDGEMENTS

The authors thank the Centro Nacional de Secuenciación Genómica- Universidad de Antioquia (Medellín, Colombia) for assistance in bioinformatic analysis and the reviewers Gabriel Yazbeck and Carlos Henrique Santos for their valuable comments that improved our manuscript.

### Funding

This work was supported by the Universidad Nacional de Colombia and Integral S. A., on 19 September 2013 and Universidad Nacional de Colombia, Sede Medellín and Empresas Públicas de Medellín, Grant CT-2013-002443-R1 "Variación genotípica y fenotípica de poblaciones de especies reófilas presentes en el área de influencia del proyecto hidroeléctrico Ituango", Grant Convenio CT-2019-000661 "Variabilidad genética de un banco de peces de los sectores medio y bajo del Río Cauca". The funders had no role in study design, data collection and analysis, decision to publish, or preparation of the manuscript.

### Grant Disclosures

The following grant information was disclosed by the authors:
Universidad Nacional de Colombia and Integral S.A., on 19 September 2013.
Universidad Nacional de Colombia, Sede Medellín and Empresas Públicas de Medellín.
Variación genotípica y fenotípica de poblaciones de especies reófilas presentes en el área de influencia del proyecto hidroeléctrico Ituango: CT-2013-002443-R1.
Variabilidad genética de un banco de peces de los sectores medio y bajo del Río Cauca: CT-2019-000661.

### Competing Interests

The authors declare that they have no competing interests.

## Author Contributions

- Ricardo M. Landínez-García conceived and designed the experiments, performed the experiments, analyzed the data, prepared figures and/or tables, authored or reviewed drafts of the paper, and approved the final draft.
- Juan Carlos Narváez conceived and designed the experiments, analyzed the data, prepared figures and/or tables, authored or reviewed drafts of the paper, and approved the final draft.
- Edna J. Márquez conceived and designed the experiments, analyzed the data, prepared figures and/or tables, authored or reviewed drafts of the paper, and approved the final draft.

## Field Study Permissions

The following information was supplied relating to field study approvals (i.e., approving body and any reference numbers):

This study analyzed muscle samples preserved in 70% ethanol, which were collected by Integral S. A., framed under an environmental permit from Ministerio de Ambiente, Vivienda y Desarrollo Territorial de Colombia # 0155 on January 30, 2009 for Ituango hydropower construction. Samples previously studied by Orozco-Berdugo & Narváez-Barandica (2014) were collected during project 111752128352 of COLCIENCIAS under collection permit #1293 of 2013 of the Universidad del Magdalena.

## DNA Deposition

The following information was supplied regarding the deposition of DNA sequences:

The *cox1* sequences are available at GenBank: MK330430–MK330494.

## Data Availability

The raw sequence contigs of 52 microsatellite loci for *Prochilodus magdalenae* are available in the Supplemental Files.

## Supplemental Information

Supplemental information for this article can be found online at http://dx.doi.org/10.7717/peerj.10327#supplemental-information.

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
