# Peer review of "Population genetics of the freshwater fish Prochilodus magdalenae (Characiformes: Prochilodontidae), using species-specific microsatellite loci"

_PeerJ, doi:10.7717/peerj.10327_

## Round 0.1 · original submission · Major Revisions

I think the manuscript indeed suffers from serious flaws.

Apart from the objections raised by the Reviewers I find the HW equilibrium analyses (e.g., l.227: how can allelic frequencies be concordant with expectations?) particularly disturbing.

Also, the interpretation of heterozygote deficiency (l.297-305 ssq) needs deep reconsideration (inbreeding vs. silent alleles and inconsistency with the explanation for the results from other authors).

I do not understand the sentence containing: showing a single allele size class in more than 50% of alleles in the studied sample (l.220-221) which, in connection with the HW note above suggest the authors confuse allele and genotype.

(High) PIC and and high polymorphism are redundant (l.136)

I do not understand the significance of the numbers at the first line of S2 Excel sheet.

·

Basic reporting

The study presents population genetics analyses on the migratory <Prochilodus magdalenae> fish species, endemic and highly conspicuous to the Magdalena-Cauca and Caribe Basins in Colombia, while presenting its first set of 11 specific microsatellite DNA markers. The study is notable for its pioneer quality and for a rather fair sample size, over 700 specimens. However, the manuscript emphasizes the use of Next-Generation Sequencing (NGS), whilst not disclosing any NGS data or even reporting on basic NGS results. These would not only be potentially helpful in a lot of predictable and unanticipated ways by the research community, but would also be consonant with the open data ethos. The text needs copy editing and verification. Specific spotted examples are highlighted directly in the annotated manuscript, along with many detailed punctual comments (signed in the PDF as "Vera").

Experimental design

The methods chosen for the development of new species-specific microsatellite markers were spot on, as there is little justification for using more traditional methods aiming this goal, currently. The population genetic analyses are also sound, pertinent and grounded on the precedent literature. Worth praising is the author's use of more current/update estimators for the genetic structure of populations, such as F'st, Jost's D
and the application of Bayesian approaches. However, Table 1 is presented in the Methods section, while reporting on new results from this original research. It should be moved to the Results section and its findings about expected, observed heterozygosity, exact p and Fis values should be reconsidered, since it has been explained it was composed of a random sample of 88 fish. Given these fish are not from the same sampling effort or location, there is little utility for these results (unless the implications of its nature is stated, accompanied of its respective meaning in discussion, since there will be a tendency towards a deficit of observed heterozygosity and departure of HWE, due to Wahlund effect, as observed in these results). Maybe it could be used to reflect and support the idea the the species does not consist in a single panmitic deme throughout the studied geographic range. Another issue worth raising awareness is the discussion of the inherent flaw in a specific feature of the Micro-Checker (MC) software, used here, presented in Waples, R.S., Journal of Heredity, 2015:106(1):1–19, doi:10.1093/jhered/esu062. The authors should reflect and consider if this could be of compromise to their intended goals ("detect potential genotyping errors") in the present study. I can't seem to locate the reporting of these results below. If it is the case, one could omit the use of this specific software, unless it supports/contradicts directly or indirectly the the study findings. If appropriate, the authors could state that the MC analyses simply did not show evidence of genotyping errors.

Validity of the findings

The paper presents a thorough description of population genetics analyses for this species, the first using species-specific microsatellite markers and will be of great value as a departing port for guiding further studies, proposing testable hypothesis about this species biology and ultimately helping in policy guidance, environmental mitigation and management decisions, in face of the coming wave of population genomic applications, fostered by NGS. This could be made explicit in the discussion (i.e. the results inherent limitations on number of markers sampled in the genome), in the context that the paper forwards 11 useful loci (~11/27 chromosomes (?), or an approximate average of less than one marker per each two different chromossomes in this species complement) and that the current study conclusions or further inquiries would greatly benefit from further future NGS based genotyping such as RAD/GBS and similar approaches, now at hand. A very positive point is that there was a strong selection criteria for loci to be applied in the genetic structure diversity analyses.

My main points to be discussed are:

1) No NGS results used for generating the data sifted to isolate microsatellites were reported (e.g. assembly results and statistics, depth of sequencing, number of contigs obtained/screened, number of microsatellites found, average PHRED quality, etc). No NGS data (e.g. FASTQ files, assemblies, etc.) was made public through adequate data repositories (e.g. NCBI's SRA). Ideally, each contig used for a validated microsatellite should be publicly revealed. Its average depth of coverage would be also a useful information. Still more preferable, the availability of alignments of short reads onto contigs for each locus should be made available or reported here. If not possible, the authors should disclose/discuss why.

2) What was the test significance (alpha value) after the Sequential Bonferroni procedure? Please, confirm if this was indeed a Holm-Bonferroni type multiple test correction or just a simple Bonferroni correction (less appropriate than the former) and clarify if it was used with all the HWE values from the whole sample, with or without the tests carried in the preliminary testing step, with the 88 fish (i.e. is there a single value for the study, or does the study show different values for Tables 1 and 2)?

3) Are the Fis values found actually accounted by breeding system endogamy? What would be the evidence supporting it, against the other concurrent hypothesis of sub-structuring and Wahlund effect? Even having the authors put it as non-excludent alternatives, in the light of the study's conclusion, I would tend to interpret these numbers as an effect of mixture of fish from different true stocks, so I advise caution with this conclusion. Same goes, in a less critical way, for disentangling effects from selection, from those of admixture.

Additional comments

The manuscript reports on a notable work in the area of Neotropical fish population genetics and adds to the expanding literature on prochilodontid species, with clear implications for production, management and conservation of an important fisheries resource and describes 11 new valid microsatellite markers isolated from <P. magdalanae>. It could be improved by a detailed copy editing (beyond this reviewer's pointed instances) and by addressing the specific questions raised, including why the NGS results were not presented and made available along the article, since they are an integral part of the works repeatability and could be used in other diverse potential works.

·

Basic reporting

no comment

Experimental design

no comment

Validity of the findings

no comment

Additional comments

This paper is effective, concise, and makes a compelling case for the study of the populations de Prochilodus magdalenae using species-specific microsatellite loci. I have no major issues with the theory or methods applied but would caution the authors to reconsider some details raised for the manuscript (see my comments below). I think it’s important to adhere to the most standardized, accessible terminology here, especially in the area of the conservation and population genetics (see my comments below). The structure of the paper is generally good, but the paragraph and sentence structure (particularly in the Introduction) needs to be better organized. While I recognize the importance of the data provided in the present study, I have detected a number of problems in the manuscript that make it unsuitable for publication the way it is. Therefore, in my opinion, there are some issues which need to be reviewed and/or answered in order for the manuscript to become suitable for publication. In addition, a review of English is required. I encourage the authors to review these issues and resubmit the manuscript for appreciation.

Line-by-line revisions:

Line 18: Change “The genetic structuring patterns...” to “The genetic structure patterns of populations....”

Line 23: Withdraw “... next-generation sequencing, bioinformatics, and...” to “... next-generation sequencing, and...”

Line 25: Withdraw “... and plausible signs of erosion...” to “... and signs of erosion...”

Line 29: Add “genetics” here: “... the genetics diversity and structure of P. Magdalenae...”

Line 39: Change “... fish species along the main river...” to “... fish species in the main river...”

Line 41-42: Change “... body sizes and high fecundities and abundances, representing around 50–80% of the biomass caught in artisanal and commercial fisheries throughout the distribution area...” to “... body sizes, high fecundities and abundances, representing around 50–80% of the biomass caught by the subsistence and commercial fisheries in some regions of your distribution area...”

Artisanal fisheries is seen as subsistence fishery. However, care should be taken when generalizing a fisheries resource as representative for all your distribuition area. The information has to be very solid.

Line 50: Add “ , ” here: “... permanent resource availability, as well as to guarantee...”

Line 54: Change “... of 2,182.67 metric tonin 2013...” to “...of 2,182.67 metric tons in 2013...”

Line 55: Add “the years of” here: “... between the years of 1978 and 2012...”

Line 56: Change “... population densities, catches (approx. 85%), and mean catch sizes...” to “... populations densities, mean catch size, and catch reduction around 85%...”

Line 61: Change “... counteract its detrimental situation...” to “...counteract this detrimental situation...”

Line 63-64: Change “... was catalogued as under critical threat in 2002 and as vulnerable since 2012 in the Colombian Red List of freshwater fishes (Mojica et al., 2012)...” to “... was catalogued as critically endangered in 2002 and, in 2012 was considered as vulnerable for the Colombian Red List of freshwater fishes (Mojica et al., 2012)...”

Line 65-66: Change “... their efforts on population reinforcements (improperly called restocking) of natural stocks in...” to “... their efforts in the restocking population of natural stocks threatened in...”

Line 69: Change "... regulation of fish farming (Povh et al...” to “... regulation of the fish farm (Povh et al...”

Line 75: Change “... for population reinforcements of natural stocks...” to “... for population restocking of natural stocks...”

Line 75: Change “Hence, natural stocks...” to “Thus, natural stocks...”

Line 81-82: Withdraw and Chance “... the observation that Prochilodus lineatus (Godoy, 1959) and Prochilodus argenteus (Godinho & Kynard, 2006) show fidelity to spawning sites (“homing”) suggests that P. magdalenae may exhibit...” to “... observations performed in the species Prochilodus lineatus (Godoy, 1959) and P. argenteus (Godinho & Kynard, 2006) show that these species present fidelity to the spawning sites (“homing”) suggesting that the species P. magdalenae can exhibit...”

Line 84-85: Withdraw and Change “... Indeed, previous genetic studies have found the population structure and/or coexistence of multiples stocks along the Magdalena River and several tributaries...” to “... Previous genetic studies have found population structure and/or coexistence of multiples stocks along the Magdalena River and of several your tributaries...”

Line 84-92: Withdraw and Change “Although this structure may result from the unregulated population reinforcements of the natural stocks, it may also reflect a natural behavior of P. magdalenae since similar patterns of genetic population structure have been found in other congeners such as Prochilodus reticulatus (López-Macías et al., 2009), P. argenteus (Hatanaka & Galetti Jr., 2003; Hatanaka, Henrique-Silva, & Galetti Jr., 2006; Barroca et al., 2012a), P. lineatus (Ramella et al., 2006; Rueda et al., 2013; Gomes et al., 2017), and Prochilodus costatus (Barroca et al., 2012a,b)” to “Although the cause this structure can be the result of the restocking of populations unregulated for the natural stocks, it can also reflect in the natural behavior of P. magdalenae since similar patterns of genetic structure have been found in other congeners such as P. reticulatus (López-Macías et al., 2009), P. argenteus (Hatanaka & Galetti Jr., 2003; Hatanaka, Henrique-Silva, & Galetti Jr., 2006; Barroca et al., 2012a), P. lineatus (Ramella et al., 2006; Rueda et al., 2013; Gomes et al., 2017), and P. costatus (Barroca et al., 2012a,b), respectively”

Line 95-97: Withdraw and Change “Likewise, we compare the genetic diversity and structure with those of five sites (Pijiño, Llanito, Mompox, Palomino, and San Marcos) previously studied by Orozco Berdugo & Narváez Barandica (2014)” to “Thus, we compare the genetic diversity and structure with the sites (Pijiño, Llanito, Mompox, Palomino, and San Marcos) previously studied by Orozco Berdugo & Narváez Barandica (2014)”

Line 98: Add “study of” here: “... their advantages in study of population genetics...”

Line 104: Change “... the river mainstream and floodplain...” to “... the river main stream and floodplain...”

Line 104-106: Add “samples, (river name?), of, farm and (fish famr name?)” here: “... a total of 725 muscle tissues samples of P. magdalenae from the river main stream (river name?) and floodplain lakes along of the different Colombian hydrographic areas of the Magdalena-Cauca and Caribe (Fig. 1; Supplementary Information) and, 40 juveniles from a local fish farm hatchery (fish farm name?).

Line 109-110: Change “... Territorial de Colombia #0155 on January 30, 2009 for Ituango hydropower...” to “... Territorial de Colombia #0155 (January 30, 2009) for Ituango hydropower...”

Line 112: Change and Add “respectively” here: “... permit #1293 of 2013 of the Universidad del Magdalena.” to “... permit #1293/2013 of the Universidad del Magdalena, respectively.”

Line 114-116: Withdraw “... P. magdalenae from the middle section of the Magdalena River was performed using the Illumina MiSeq v.2 instrument using the "whole genome shotgun" strategy and the Nextera library preparation kits for the sequence...” to “... P. magdalenae for the region of middle Magdalena River was performed using the Illumina MiSeq v.2 platform (manufacturer, city, country). An alternative approach to shotgun whole genome sequencing (WGS) was applied using the Nextera Library preparation kit (specify the Nextera kit) (manufacturer, city, country) for the sequence...”

Line 117: Change “... steps concerning the read cleaning, contig assemblage, identification...” to “... steps regarding the reads quality and filtering, contig assemblage, identification...”

Line 120-126: Redo all this passage (see my comments on this topic – Major (1)).

Line 137-139: Withdraw and Change “... using LIZ500 (Applied Biosystems) as the internal molecular size. Allelic fragments were denoted according to their molecular size and scored using GeneMapper v.4.0 (Applied Biosystems)...” to “... using the GeneScan LIZ-500 standad size (Applied Biosystems, city, country) to determine fragment length. The alleles were scored based on the consistent pattern of their stutter peaks, and on the peak intensity corresponding to each individual at each locus using GeneMapper v4.0 (Applied Biosystems, city, country)...”

Line 142-144: Withdraw and Change “... for departures from Hardy–Weinberg linkage equilibria as well as the observed (HO) and expected (HE) heterozygosity and the inbreeding coefficient (FIS) were estimated using Arlequin v.3.5.2.2 (Excoffier, Laval, & Schneider...” to “... for Hardy–Weinberg equilibrium (HWE), observed (HO) and expected heterozygosity (HE) and, the inbreeding coefficient (FIS) were estimated using Arlequin v3.5.2.2 software (Excoffier, Laval, & Schneider...”

Line 146-147: Withdraw and Change “... for each marker were calculated with GenAlEx v.6.503 (Peakall & Smouse, 2006) and Cervus v.3.0.7 (Marshall et al., 1998), respectively” to ... for each SSR were calculated with GenAlEx v6.503 (Peakall & Smouse, 2006) and Cervus v3.0.7 software (Marshall et al., 1998), as well as to estimate the genetic diversity of P. magdalenae”.

Line 148-150: Withdraw “The average number of alleles per locus, observed and expected average heterozygosities, and fixation index (Hartl & Clark, 1997) were calculated with GenAlex v.6.503 (Peakall & Smouse, 2006) to estimate the genetic diversity of P. magdalenae”.

Line 150-153: Withdraw and Change “... geographical samples was calculated using the standardized statistics F ́ST (Meirmans, 2006) and Jost ́s Dest (Meirmans & Hedrick, 2011) and analysis of molecular variance (AMOVA) (Meirmans, 2006) with 10,000 permutations and bootstraps included in GenAlex v.6.503...” to “... geographical samples were calculated using the standardized statistics F ́ST (Meirmans, 2006) and Jost ́s Dest (Meirmans & Hedrick, 2011) and, analysis of molecular variance (AMOVA) (Meirmans, 2006) with 10,000 permutations and bootstraps included in GenAlex v6.503 software...”.

Line 156-167: Add "software" and Withdraw “ . ” for all programs used. Example: Structure v.2.3.4 to Structure v2.3.4 software
CLUMPP v.1.1.2b to CLUMPP v1.1.2b software
Distruct v.1.1 to Distruct v1.1 software

Line 176: Withdraw and Change “... using the software BayeScan v.2.1 (Foll & Gaggiotti...” to “... using the BayeScan v2.1 software (Foll & Gaggiotti...”

Line 181-192: Add "software" and Withdraw “ . ” for all programs used. Example:
software jModelTest to jModelTest software
software MrBayes v.3.2.6 to MrBayes v3.2.6 software
Figtree v.1.4.3 to Figtree v1.4.3 software

Line 208-209: Change “... revealed that 8 of 11 loci exhibit allelic frequencies concordant with Hardy-Weinberg equilibrium expectations in at least one case (Table...” to ... revealed that 7 of 11 loci exhibit allelic frequencies concordant with Hardy-Weinberg equilibrium expectations in at least one case (see Table...”.

Line 225: Withdraw “... magnitude, heterozygosity deficits and inbreeding coefficients...” to “... magnitude and inbreeding coefficients...”

Line 228-230: Change “... of the tests performed using Bottleneck (Table 4) were significant for all populations under the infinite alleles model (IAM) and for most populations under the two-phase model (TPM), whereas they were generally non-significant under the stepwise mutation model (SMM)” to “... of the genetic bottleneck were significant for all populations under the infinite alleles model (IAM) and for most populations under the two-phase model (TPM), however, was non-significant for the stepwise mutation model (SMM) (Table 4)”.

Line 241: Change “... and AMOVA (F ́ST(7, 1407) = 0.009...” to “...and AMOVA (F ́ST(7, 1407) = 0.009...”

Line 242-244: Change “... statistics F ́ST (Meirmans, 2006) and Jost ́s Dest (Meirmans, & Hedrick, 2011) showed additional genetic differences among Atrato, the fish hatchery, Sinú, and the remaining rivers (Table 6) as well as...” to “... statistics F ́ST (Meirmans, 2006) and Jost ́s Dest (Meirmans, & Hedrick, 2011) showed additional genetic differences among Atrato, the fish farm hatchery, Sinú, and the remaining rivers (Table 6), as well as...”

Major:
(1) The authors discuss the parameters required to validate new microsatellite (SSR) primers, but it is not clear what these parameters. The criteria adopted by the authors to choose the 11 microsatellite loci, only two I consider important for validation of new microsatellite loci: (i) value of F or FIS (however, this depends directly on the pValue for the deviation of the Hardy-Weinberg equilibrium - HWE ) and (ii) the Polymorphic Information Content (PIC) - loci should be polymorphic. The authors speak of low levels of heterozygosity deficit, but when I see F or FIS values (see Table 1, HO < HE), I observe high levels of heterozygosity deficiency for all loci, as well as 16 loci with deviation HWE after Bonferroni correction (5%, p <= 0.05 / 21 = 0.00238). In addition, it is important to perform the linkage disequilibrium (LD) test for the validation of the 21 SSR loci. I suggest you take the LD test and add the information in the manuscript.

1.1. What are the criteria actually used for the choice of the 11 microsatellite loci for the present study? “I am not against the choice of 11 loci for this study, however, I would like this to be clear to me”.

1.2. What did mean by low levels of heterozygosity deficit?

1.3. Why did not check the private alleles in the populations studied? I suggest adding the information from the private alleles in the manuscript.

(2) Why was not an analysis for isolation by distance (IBD) in the distribution area of the species studied? This would be interesting and would greatly contribute to the manuscript data. I suggest that this analysis be carried out.

(3) I suggest the author better explain of the data information found in Figs. 2, 3 and 4 (structure), as well as the DAPC data. The current form as presented is very fragmented and can generate misinterpretation.

What I observed in this part of the manuscript related to the structure and DAPC data:
Figure 2A, two populations on the Magdalena River, and Fig. 2C two populations - Sinú and Atrato Rivers (Population 1) and the Cauca, Magdalena, Cesar, San Jorge and Nare Rivers (Population 2), respectively. The Fig. 3A shows that Magdalena (MG), Cesar (CS), San Jorge (SJ) and Nare (NA) rivers populations are more related than wtih the Cauca (CA) river populations. This is clearer with Fig. 3B and 3C, when we observed the Cauca (CA) and Magdalena (MG) rivers populations separately. Individuals from the fish farming hatchery, the author suggests that they originate from several rivers. It would be important for the authors to have this information about the origin of the individuals fish farm. Data on distance isolation (IBD) and a UPGMA tree would help to better understand Figs. 2, 3 and 4.

NOTE: I suggest the authors make a new run in the strucuture software up to a K = 8 (seven rivers + fish farm), adding each result of K to a single figure. Present the data for gene flow between the studied rivers. This information will be important to verify the level of reproductive isolation that these populations present.

Even with the existence of structure, the DAPC data show a genetic mix between the studied rivers, but this would be justified by the repopulation carried out in the region with individuals of the species Prochilodus magdalenae. The authors constructed a Bayesian tree (Fig. 5) using the cox1 gene. I suggest building a new tree containing the origin of each sample to understand the DAPC data.

(4) When checking Tables 2 and 3, it is observed that the majority of the populations of the target species are with values above 10% for FIS values. The FST values (Table 6) suggest that there is a good gene flow among the populations, however, the FIS values indicate the occurrence of mating with related individuals, what can be caused by the genetics bottleneck and/or restocking causing by the mixture of the population (the parents used in the restocking program probably originate from the same area of study). In addition, possible signs of local adaptation could have been verified (from high FST values, number of private and low alleles or absence of genetic flow).

The Wahlund effect on large rivers with the fragmentation of populations may be due to the construction of hydroelectric power plants (barrier to gene flow) e/or restocking program consequence performed in the rivers (different populations coexisting). To confirm the Wahlund effect due to the coexistence of genetic stocks in the study area, it will be important for the author to see the allelic frequencies that are different between populations and which has caused the heterozygosity deficit.

“Thus, it is lacking in the discussion a greater exploration of the consequences that this can bring if the management for species is not applied in the studied área”.

Minor:
(1) In the methodology nothing was found about of the DNA extraction step in the P. magdalenae samples. It is important that this information is contained in the material and methods. Please, add in the body of the text the method applied for DNA extraction.

(2) “The extension step and a final elongation were absent in this thermal profile”. Justify why the absence of this step in the PCR? Did you use any method on this? If yes, it should be cited in the text.

(3) Caution: all chemicals and equipment must bear the following information - manufacturer, city and country. Example: GeneScan Liz-500 (-250) standard size (Applied Biosystems, Waltham, USA) or 96-well Veriti™ Thermal Cycler (Applied Biosystems, Waltham, USA)

(4) I am does not understand what the author meant by this passage - Line 197-198: "A total of 21 of the 50 loci microsatellite evaluated were polymorphic and showed allelic frequencies that departed from Hardy-Weinberg equilibrium".

How do you know that the allelic frequency departed from Hardy-Weinberg equilibrium? I suggest that add a supplementary table with the data of the allelic frequency.

(5) Standardize in the manuscript: “Ho” to “HO”, “He” to “HE”, “F’ST” to “F’ST” and “F” to “FIS”, respectively.

(6) The fixation index (F) and inbreeding coefficient (FIS) they are not the same thing? Review table 3.

(7) The genetic structure tends to decrease when populations are mixed, increasing or restoring the gene flow among individuals of different populations. Thus, I did not understand what the author wanted to say in the line 275-276: "... the genetic structure of the samples shaped by the mixture of two genetic stocks..."

---

## Round 0.2 · Major Revisions

Some major issues remain to be solved, besides those presented by the reviewers:
- I could not find data/results regarding the validation of the claimed species specificity of the markers; in case this was not properly assessed, as I fear, the claim should be omitted from the title and throughout the text.
- I could not find any details on the taxonomic identification of the samples in M&M section.
- My reservations on HW analyses remain; in particular: how can you justify to have selected loci that «showed Hardy-Weinberg disequilibrium» (lines 226-227) and «the analysis across loci showed significant departures from Hardy-Weinberg equilibrium expectations in all rivers evaluated» (l. 238-9)? In the absence of a comparison of observed and expected genotype distributions (per maker and population) it is impossible to infer the possible causes
- I think when the «deficit of heterozygosity» (e.g.., l. 251) is mentioned, it should correctly be stated as «deficit of observed heterozygotes».
- Phylogeny (based on mtDNA only, which means reflecting just maternal lineages) is also troubling: how come that some P. reticulatus are clustered with P. magdalenae (and that «genetic distances (Table S7) were larger among haplotypes of P. magdalenae (0.002 – 0.014) than among P. magdalenae and P. reticulatus haplotypes (0.002 – 0.010).»? This casts serious doubts on the species assignment among this group (N.B: in connection with my first criticism on the claimed specificity. ). Below it is stated that there is a «proposal that P. magdalenae and P. reticulatus represent only one species». How to conciliate/solve this issue? Hybridiztion?
- I disagree with the conclusion: «both heterologous and species-specific microsatellite loci revealed a general deficit of heterozygotes in all samples, indicating that its causes are biological rather than technical.» In fact the Authors list a series of problems for the so-called ‘heterologous’ microsatellites which also apply to the claimed ‘species.-specific’. I think my previous objections were not removed, as I could not find any specific approach to elucidation of the cause of the deficit (inbreeding /substructure vs. silent alleles/technical problems).
Minor corrections
- Please confirm if what is meant by «Results of the genetic bottleneck (Table 4)» is: Results of the genetic bottleneck tests; if so, correct accordingly.
- In the supplemental file «Raw sequence contigs, longitude and some identification parameters of 52 microsatellite loci selected for Prochilodus magdalenae» by ‘longitude’ you mean ‘length’? If so, modify accordingly.
Finally, since - as he Authors mention - there is protective legislation regarding this animal, forensic cases may arise, for the solution of which this type of markers may be extremely valuable, provided they are validated. Accordingly, I advise the Authors to explicitly consider this application, consulting the relevant literature, namely from specialized journals such as Forensic Science International Genetics (and the Supplement Series), where relevant papers were recently published and recommendations/guidelines on animal forensic genetics issued.

·

Basic reporting

The authors have full, direct and satisfactorily addressed all remarks, questioning and suggestions I have made in the first round of revision. They have added a profuse and detailed suite of supplemental data that will certainly enrich the paper’s open virtual environment presentation. I also take the opportunity to thank back the authors for their acknowledgement and I hope they will forgive me, though, and cope with new little remarks and details I have freshly raised in this second round or review, of elements already present in the original manuscript, along one main important point:
-Although there are no rules of name translation, throughout the text, tables and supplemental material, do prefer the use of the English term “the Caribbean” instead of “Caribe” (e.g. L18, L94 and Supplemental Table 1);
- Line 92 Since the objective deals with an atemporal aspect of the study (what it does) prefer the used of the present tense instead the past tense: "this study tests the hypothesis";

Experimental design

No new comments;

Validity of the findings

No new comments;

Additional comments

My only point of concern is that the paper devalues itself while not making clear in the discussion and in the conclusion that these are indeed the FIRST species-specific microsatellite DNA markers published for this fish species (see instances on the annotated review manuscript), at least to the best of my knowledge. A search of GenBank revealed a dubious claim of 17 P. magdalenae species-specific microsatellites (implicit in the title of an unpublished manuscript – e.g. entries MH586829 through MH586842) for what actually seem to be other already published microsatellite loci for P. lineatus, P. argenteus and P. costatus. If the authors arrive to the conclusion this here is the first set of microsatellite markers for P. magdalenae, it certainly needs to be brought to light in the final work.

·

Basic reporting

The present study shows great improvements after the suggested revisions and has relevant information for the conservation of the species Prochilodus magdalenae. The information presented here is clear and the article presents an adequate structure. Figures and tables (supplementary material) bring information that strengthens the content presented in the paper. The references are adequate for the discussion of the results, however, I felt a lack of more current references from the last 3 years (2017-2020). The paper presents only 7.8% of references for the years 2017-2020.

Experimental design

The research presented here is adequate to the scope and objective of the journal. The paper's objective is well defined and relevant to the study area. The methodology applied is adequate and describes in detail the study.

Validity of the findings

The underlying data has been provided (supplementary material) and they are robust for a better understanding of the results. The conclusions are very clear from the study proposal and linked to the research question.

Additional comments

I congratulate the authors for the importance of the study and its contribution to the management and conservation of the species Prochilodus magdalenae. It would be interesting if the authors had applied a greater number of microsatellite loci, as well as some mitochondrial genes for further investigation. However, the work brings relevant information and contribution to the study area. I highlight here the sample size to the number of microsatellite markers used in the study. The review of the paper brought improvements to the study.

---

## Round 0.3 · accepted · Accept

Although the analysis of departure from HWE is still weak and the forensic implications poorly addressed, I could not find major errors which would prevent publication.

·

Basic reporting

The authors have, once more, addressed all suggestions and points raised. In particular, regarding one of my inquiries, they have argued for their position on not emphasizing the novelty aspect of the developed microsatellite markers set. They have opted for a more humble stance, conceding to the benefit of the doubt, relating to a previous 2018 GenBank deposit of microsatellite clones for P. magdalenae, by a third party. While, as the own efforts of the present study might testify, there is a huge gap between isolating/characterizing clones and empirically validating it for practical use, and while a peer-review publication should take precedence over isolated GenBank accessions as a primary source, given the nature of the study, which focus on its genetic diversity structure description goal, I am ready to acknowledge this position, knowing the molecular markers developed by the paper’s team will certainly be readily available to other research initiatives aiming this species, all the same. It is an important fisheries resource and, from a biological perspective, it clearly needs more study and an integrated taxonomic revision, mainly in regard to its status along with P. reticulans, as hinted in this report and on previous literature and also discussed during this peer-review process. I think the paper is suitable for publication in its current state and I congratulate its authors and research institutions for their efforts.

Experimental design

No new comments.

Validity of the findings

No new comments.

Additional comments

No new comments.

·

Basic reporting

All questions were resolved and improved in the body of the paper.

Experimental design

All questions were resolved and improved in the body of the paper.

Validity of the findings

All questions were resolved and improved in the body of the paper.

Additional comments

The authors made the improvements suggested in the final version, and justified all the points covered in the review. I consider the paper adequate and ready to be published in Peerj.